# Neuronal hyperexcitability is a DLK-dependent trigger of herpes simplex virus reactivation that can be induced by IL-1

Sean R Cuddy[1,2†], Austin R Schinlever[1†], Sara Dochnal[1], Philip V Seegren[3], Jon Suzich[1], Parijat Kundu[1], Taylor K Downs[3], Mina Farah[1], Bimal N Desai[3], Chris Boutell[4], Anna R Cliffe[1]*

[1]Department of Microbiology, Immunology and Cancer Biology, University of Virginia, Charlottesville, United States; [2]Neuroscience Graduate Program, University of Virginia, Charlottesville, United States; [3]Department of Pharmacology, University of Virginia, Charlottesville, United States; [4]MRC-University of Glasgow Centre for Virus Research (CVR), Garscube Campus, Glasgow, United Kingdom

**Abstract** Herpes simplex virus-1 (HSV-1) establishes a latent infection in neurons and periodically reactivates to cause disease. The stimuli that trigger HSV-1 reactivation have not been fully elucidated. We demonstrate HSV-1 reactivation from latently infected mouse neurons induced by forskolin requires neuronal excitation. Stimuli that directly induce neurons to become hyperexcitable also induced HSV-1 reactivation. Forskolin-induced reactivation was dependent on the neuronal pathway of DLK/JNK activation and included an initial wave of viral gene expression that was independent of histone demethylase activity and linked to histone phosphorylation. IL-1β is released under conditions of stress, fever and UV exposure of the epidermis; all known triggers of clinical HSV reactivation. We found that IL-1β induced histone phosphorylation and increased the excitation in sympathetic neurons. Importantly, IL-1β triggered HSV-1 reactivation, which was dependent on DLK and neuronal excitability. Thus, HSV-1 co-opts an innate immune pathway resulting from IL-1 stimulation of neurons to induce reactivation.

*For correspondence: cliffe@virginia.edu

†These authors contributed equally to this work

Competing interests: The authors declare that no competing interests exist.

## Introduction

Herpes simplex virus-1 (HSV-1) is a ubiquitous human pathogen that is present in approximately 40–90% of the population worldwide (*Arvin, 2007*). HSV-1 persists for life in the form of a latent infection in neurons, with intermittent episodes of reactivation. Reactivation from a latent infection and subsequent replication of the virus can cause substantial disease including oral and genital ulcers, herpes keratitis, and encephalitis. In addition, multiple studies have linked persistent HSV-1 infection to the progression of Alzheimer's disease (*Itzhaki, 2018*). Stimuli in humans that are linked to clinical HSV-1 reactivation include exposure to UV light, psychological stress, fever, and changes in hormone levels (*Suzich and Cliffe, 2018*). How these triggers result in reactivation of latent HSV-1 infection is not fully understood.

During a latent infection of neurons, there is evidence that the viral genome is assembled into a nucleosomal structure by associating with cellular histone proteins (*Deshmane and Fraser, 1989*). The viral lytic promoters have modifications that are characteristic of silent heterochromatin (histone H3 di- and tri-methyl lysine 9; H3K9me2/3, and H3K27me3) (*Wang et al., 2005*; *Knipe and Cliffe, 2008*; *Cliffe et al., 2009*; *Kwiatkowski et al., 2009*), which is thought to maintain long-term silencing of the viral lytic genes. Hence, for reactivation to occur, viral lytic gene expression is induced from promoters that are assembled into heterochromatin and in the absence of viral proteins, such as VP16, which are important for lytic gene expression upon de novo infection and full reactivation

(*Thompson et al., 2009*; *Kim et al., 2012*). The initiation of viral lytic gene expression, including *VP16*, during reactivation is therefore dependent on host proteins and the activation of cellular signaling pathways (*Suzich and Cliffe, 2018*). However, the full nature of the stimuli that can act on neurons to trigger reactivation and the mechanisms by which expression of the lytic genes occurs have not been elucidated.

One of the best characterized stimuli of HSV reactivation in primary neuronal models is nerve growth factor (NGF) deprivation and subsequent loss of PI3K/AKT activity (*Wilcox and Johnson, 1988*; *Wilcox et al., 1990*; *Camarena et al., 2010*). Previously, we found that activation of the c-Jun N-terminal kinase (JNK) cell stress response via activation of dual leucine zipper kinase (DLK) was required for reactivation in response to loss of NGF signaling (*Cliffe et al., 2015*). In addition, recent work has identified a role for JNK in HSV reactivation following perturbation of the DNA damage/repair pathways, which also triggers reactivation via inhibition of AKT activity (*Hu et al., 2019*). DLK is a master regulator of the neuronal stress response, and its activation can result in cell death, axon pruning, axon regeneration or axon degeneration depending on the nature of activating trigger (*Tedeschi and Bradke, 2013*; *Geden and Deshmukh, 2016*). Therefore, it appears that HSV has co-opted this neuronal stress pathway of JNK activation by DLK to induce reactivation. One mechanism by which JNK functions to promote lytic gene expression is via a histone phosphorylation on S10 of histone H3 (*Cliffe et al., 2015*). JNK-dependent histone phosphorylation occurs on histone H3 that maintains K9 methylation and is therefore known as a histone methyl/phospho switch, which likely permits viral lytic gene transcription without the requirement for recruitment of histone demethylases (*Fischle et al., 2005*; *Gehani et al., 2010*). This initial wave of viral lytic gene expression is known as Phase I, and also occurs independently of the lytic transactivator VP16. In addition, late gene expression in Phase I occurs independent of viral genome replication (*Kim et al., 2012*; *Cliffe and Wilson, 2017*). A sub-population of neurons then progress to full reactivation (also known as Phase II), which occurs 48–72 hr post-stimulus and requires both VP16 and histone demethylase activity (*Cliffe et al., 2015*; *Liang et al., 2009*; *Liang et al., 2013*; *Messer et al., 2015*; *Hill et al., 2014*), and includes viral DNA replication. However, this bi-phasic progression has not been observed in some models of reactivation such as axotomy, which results in more rapid viral gene expression and a dependence on histone demethylase activity for viral gene expression at the earliest time points investigated (*Liang et al., 2009*).

The aim of this study was to determine if we could identify novel triggers of HSV reactivation and determine if they involved a bi-phasic mode of reactivation. We turned our attention to forskolin treatment and neuronal hyperexcitability because hyperstimulation of cortical neurons following forskolin treatment or potassium chloride mediated depolarization has previously been found to result in a global histone methyl/phospho switch (*Noh et al., 2015*). Whether this same methyl/phospho switch occurs in different types of neurons, including sympathetic neurons, is not known. Although forskolin has previously been found to induce HSV reactivation, (*Smith et al., 1992*; *Colgin et al., 2001*; *De Regge et al., 2010*; *Danaher et al., 2003*), the mechanism by which forskolin induces reactivation is not known. In particular, it is unknown if forskolin acts via causing increased neuronal activity and/or as a consequence of activation of alternative cAMP-responsive proteins including PKA and CREB. Hyperexcitability of neurons is correlated with changes in cellular gene expression, increased DNA damage (*Alt and Schwer, 2018*; *Madabhushi et al., 2015*), and epigenetic changes including H3 phosphorylation (*Noh et al., 2015*). However, DLK-mediated activation of JNK has not been linked to changes in cellular gene expression nor epigenetic changes in response to hyperexcitability. Using a variety of small-molecule inhibitors, we found that forskolin-induced reactivation was dependent on ion-channel activity. In support of a role for neuronal hyperexcitability causing HSV reactivation, stimuli that are well established as causing heightened neuronal activity also induced HSV to undergo reactivation. In addition, DLK and JNK activity were required for an initial wave of viral lytic gene expression, which occurred prior to viral DNA replication and independently of histone demethylase activity, indicating that hyperstimulation-induced reactivation also involves a biphasic viral gene expression program.

We were also keen to determine whether we could identify a physiological stimulus for HSV reactivation that acts via causing neurons to enter a hyperexcitable state. IL-1β is released under conditions of psychological stress and fever (*Ericsson et al., 1994*; *Goshen and Yirmiya, 2009*; *Koo and Duman, 2009*; *Saper and Breder, 1994*); both known triggers of clinical HSV reactivation (*Glaser and Kiecolt-Glaser, 1997*; *Cohen et al., 1999*; *Chida and Mao, 2009*). IL-1β has previously

been found to induce heightened neuronal activity (*Vezzani and Viviani, 2015*; *Schneider et al., 1998*; *Binshtok et al., 2008*). However, an intriguing feature of IL-1β signaling is its ability to have differential effects on different cell types. For example, IL-1β is involved in the extrinsic immune response to infection via activation of neutrophils and lymphocytes (*Sims and Smith, 2010*). In addition, it can act on non-immune cells including fibroblasts to initiate an antiviral response (*Orzalli et al., 2018*; *Aarreberg et al., 2019*), as has previously been described for lytic infection with HSV-1 (*Orzalli et al., 2018*). Given these differential downstream responses to IL-1β signaling, we were particularly interested in the effects of IL-1β treatment of latently-infected neurons. Interestingly, we found that IL-1β was capable of inducing reactivation of HSV from mature sympathetic neurons. Inhibition of voltage-gated sodium and hyperpolarization-activated cyclic nucleotide-gated (HCN) channels impeded reactivation mediated by both forskolin and IL-1β. Activity of the cell stress protein DLK was also essential for IL-1β-mediated reactivation. We therefore identify IL-1β as a novel trigger from HSV reactivation that acts via neuronal hyperexcitability and highlight the central role of JNK activation by DLK in HSV reactivation.

## Results

### Increased intracellular levels of cAMP induces reactivation of HSV from latent infection in murine sympathetic neurons

Both forskolin and cAMP mimetics are known to induce neuronal hyperexcitation and have previously also been found to trigger HSV reactivation (*Smith et al., 1992*; *Colgin et al., 2001*; *De Regge et al., 2010*; *Danaher et al., 2003*). Using a model of HSV latency in mouse sympathetic neurons isolated from the superior-cervical ganglia (SCG) (*Cliffe et al., 2015*) we investigated whether forskolin treatment induced reactivation in this system and the potential mechanism resulting in the initial induction of viral lytic gene expression. Sympathetic SCG neurons were infected with a Us11-GFP tagged HSV-1 (*Benboudjema et al., 2003*) at a multiplicity of infection (MOI) of 7.5 PFU/cell in the presence of acyclovir (ACV). After 6 days the ACV was washed out and the neuronal cultures monitored to ensure that no GFP-positive neurons were present. Two days later, reactivation was triggered by addition of forskolin (*Figure 1A*). WAY150138 was added to the media post-reactivation to prevent cell-to-cell spread (*Newcomb and Brown, 2002*). As represented in *Figure 1B*, forskolin can act either extracellularly on ion channels or intracellularly to activate adenylate cyclase (*Hoshi et al., 1988*; *Kandel, 2012*; *de Rooij et al., 2000*). Activation of adenylate cyclase results in the propagation of second-messenger pathways resulting from activation of PKA, EPAC1 (Exchange Factor directly Activated by cAMP, also known as Rap Guanine Nucleoside Exchange Factor 1) or EPAC 2. In addition, cAMP can act directly on cyclic nucleotide-gated ion channels, and PKA can also modulate ion-channel activity via phosphorylation. Dideoxy-forskolin (dd-forskolin) is a cell-impermeable forskolin analog that can act directly on voltage-gated ion channels but does not activate adenylate cyclase (*Hoshi et al., 1988*; *Gandía et al., 1997*). We found addition of forskolin but not dd-forskolin triggered robust HSV reactivation (*Figure 1C*). A slight increase in GFP-positive neurons did occur with dd-forskolin treatment compared to mock (approximately 6.5-fold increase compared to a 130-fold increase for forskolin). Based on a Tukey's multiple comparison test, this change from mock treated neurons was not significant (p=0.07), however, a direct comparison between mock and dd-forskolin using a T-test suggested a significant induction (p=0.03). Therefore, direct stimulation of ion-channels by dd-forskolin may trigger some reactivation. However, maximal reactivation requires forskolin to enter neurons. Treatment of latently-infected primary neurons with a cAMP mimetic (8-bromo-cAMP) was sufficient to trigger reactivation (*Figure 1D*), suggesting that increased intracellular levels of cAMP are capable of inducing HSV reactivation. Furthermore, inhibition of adenylate cyclase activity using SQ22, 536 (*Haslam et al., 1978*) significantly diminished HSV reactivation (*Figure 1E*). Therefore, activation of adenylate cyclase, which results in increased intracellular levels of cAMP, is required for robust forskolin-mediated reactivation.

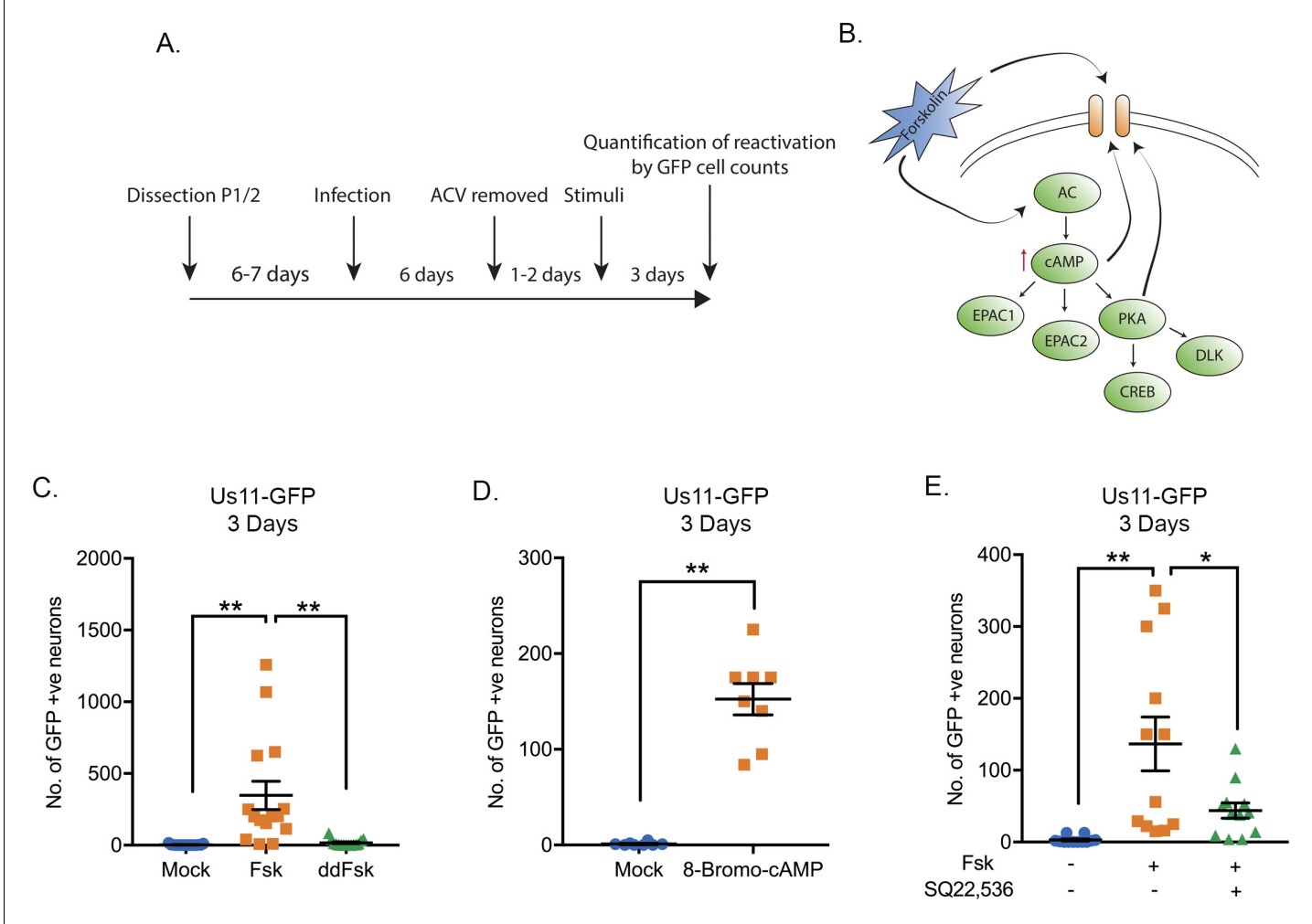

**Figure 1.** HSV-1 Reactivation from sympathetic neurons is induced by adenylate cyclase activation. (A) Schematic of the primary sympathetic superior cervical ganglia (SCG)-derived model of HSV latency. Reactivation was quantified based on Us11-GFP-positive neurons in presence of WAY-150168, which prevents cell-to-cell spread. (B) Schematic of the cellular pathways activated by forskolin treatment. Forskolin can act both intracellularly to activate adenylate cyclase (AC) and increasing the levels of cAMP or extracellularly on ion channels. (C) Numbers of Us11-GFP-positive neurons following addition of either forskolin (60 µM) or cell-impermeable dideoxy-forskolin (60 µM) treatment of latently-infected sympathetic neurons. (D) Numbers of Us11-GFP-positive neurons following treatment with a cAMP mimetic 8-Bromo-cAMP (125 µM). (E) Reactivation, quantified by Us11-GFP-positive neurons, was induced by forskolin in the presence or absence of the adenylate cyclase inhibitor SQ22,536 (50 µM). In C-E each point represents a single biological replicate, and the mean and standard errors of the mean (SEM) are also shown. In D statistical comparisons were made using an unpaired t-test. In C and E statistical comparisons were made using a one-way ANOVA with a Tukey's multiple comparisons test. *p<0.05, **p<0.01. The online version of this article includes the following source data for figure 1:

**Source data 1.** Quantification of GFP-positive neurons for *Figure 1*.

## DLK and JNK activity are required for the early phase of viral gene expression in response to forskolin treatment

We previously found that DLK-mediated JNK activation was essential for Phase I reactivation following interruption of nerve growth factor signaling (*Cliffe et al., 2015*). To determine whether DLK and JNK activation were crucial for reactivation in response to forskolin, neurons were reactivated in the presence of the JNK inhibitor SP600125 (*Figure 2A*) or the DLK inhibitor GNE-3511 (*Patel et al., 2015*; *Figure 2B*). Because DLK has been proposed as a target to prevent neuronal cell death or axon degeneration in neurological disease, GNE-3511 was recently developed as a small-molecule inhibitor of DLK that shows selective inhibition of DLK activity and protection against axon pruning with an IC50 of 0.1 µM (*Patel et al., 2015*). Both the JNK and DLK inhibitors prevented forskolin-

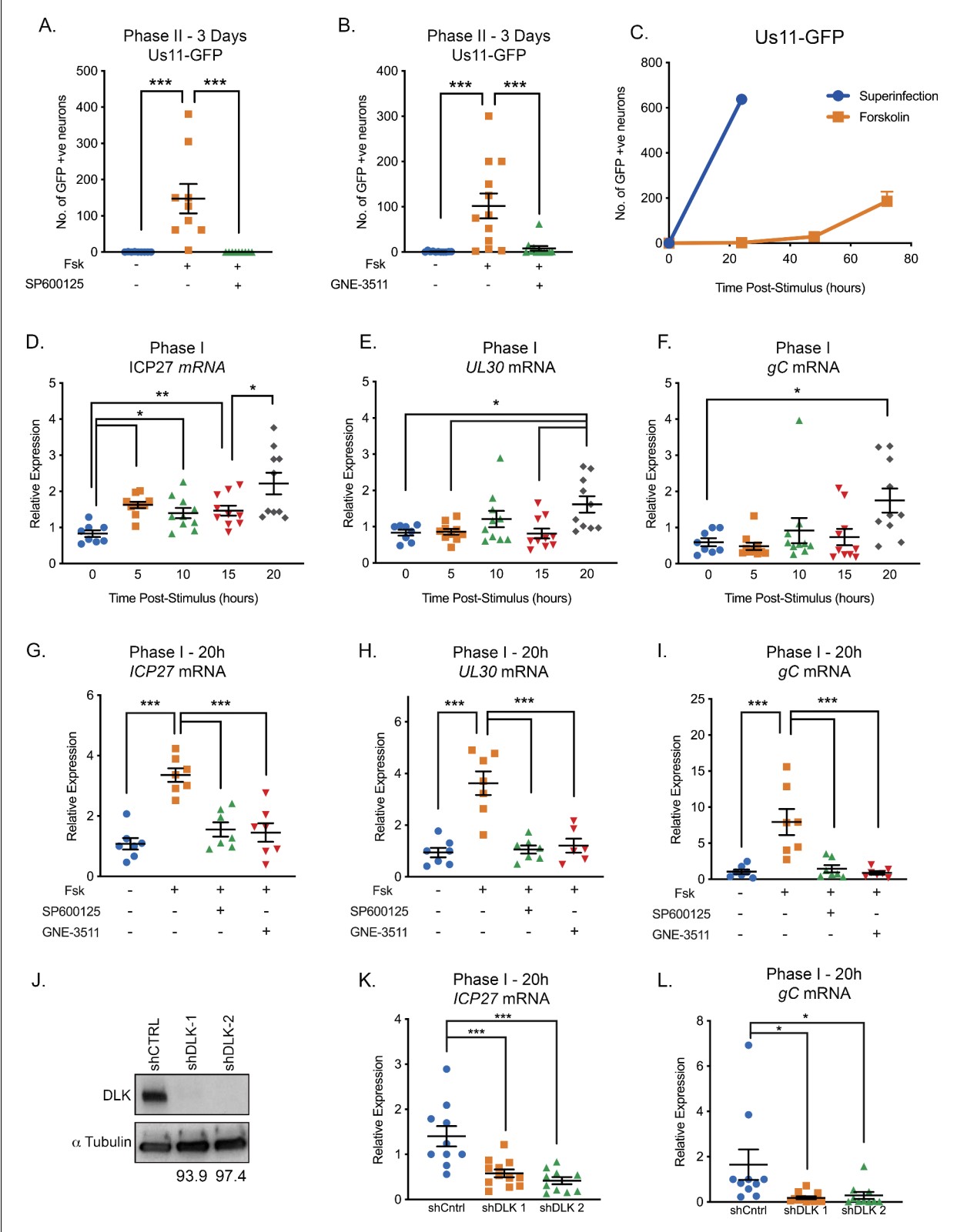

**Figure 2.** Reactivation triggered by forskolin involves a DLK/JNK-dependent phase I of viral gene expression. (A) Reactivation was induced by forskolin in the presence of JNK inhibitor SP600125 (20 μM). (B) Reactivation was induced by forskolin in the presence of the DLK inhibitor GNE-3511 (4 μM). In A and B each experimental replicate is shown. (C) Reactivation was induced by forskolin or superinfection with a wild-type (F strain) HSV-1 (MOI of 10 PFU/cell) and qualified based on Us11-GFP-positive neurons (n = 3). (D–F) RT-qPCR for viral mRNA transcripts following forskolin treatment of latently

*Figure 2 continued on next page*

*Figure 2 continued*

infected SCGs. (**G–I**) RT-qPCR for viral lytic transcripts at 20 hr post-forskolin treatment and in presence of the JNK inhibitor SP600125 (20 µM) and the DLK inhibitor GNE-3511 (4 µM). (**J**) Neurons were transduced with a non-targeting shRNA control lentivirus or two independent lentiviruses expressing shRNAs that target DLK (shDLK-1, shDLK-2). Western-blotting for DLK or β-III tubulin was carried out 3 days post transduction. The percentage knockdown of DLK normalized to β-III tubulin is shown. (**K and L**) RT-qPCR for viral mRNA transcripts following forskolin treatment of latently infected SCGs that were either transduced with the shRNA control or shRNA DLK lentiviruses. In D-I, K, and L, each experimental replicate is represented. Statistical comparisons were made using a one-way ANOVA with a Tukey's multiple comparison. $*p<0.05$, $**p<0.01$, $***p<0.001$. The mean and SEM are shown. The online version of this article includes the following source data and figure supplement(s) for figure 2:

**Source data 1.** Quantification of GFP-positive neurons, RT-qPCR and western blot band densities for *Figure 2*.
**Figure supplement 1.** Reactivation triggered by forskolin triggers a wave of lytic gene expression that precedes DNA Replication and infectious virus production.
**Figure supplement 1—source data 1.** Quantification of HSV titer, GFP-positive neurons and RT-qPCR for *Figure 2—figure supplement 1*.
**Figure supplement 2.** Effect of PKA, CREB, Rapgef2 and EPAC Inhibition on HSV-1 Reactivation.
**Figure supplement 2—source data 1.** Quantification of GFP-positive neurons and RT-qPCR for *Figure 2—figure supplement 2*.

mediated reactivation based on the number of GFP-positive neurons at 3 days post-stimulus. These data therefore indicate forskolin-mediated reactivation is dependent on the neuronal stress pathway mediated by DLK activation of JNK.

Because we and others previously found that JNK activation results in a unique wave of viral gene expression in response to inhibition of nerve growth factor signaling (*Kim et al., 2012*; *Cliffe et al., 2015*; *Cliffe and Wilson, 2017*), we were especially intrigued to determine whether forskolin triggers a similar wave of JNK-dependent viral gene expression. The previously described bi-phasic progression to viral reactivation is characterized by viral DNA replication and production of infectious virus, occurring around 48–72 hr post-stimulus (*Kim et al., 2012*), but with an earlier wave of lytic gene expression occurring around 20 hr post-stimulus. To determine whether forskolin-mediated reactivation results in a similar kinetics of reactivation, we investigated the timing of Us11-GFP synthesis, viral DNA replication, production of infectious virus, and lytic gene induction following forskolin treatment. In response to forskolin treatment, Us11-GFP synthesis in neurons started to appear around 48 hr post-reactivation, with more robust reactivation observed at 72 hr (*Figure 2C*). In contrast to forskolin-mediated reactivation, the number of GFP-positive neurons following superinfection with a replication competent wild-type virus resulted in a rapid induction of GFP-positive neurons by 24 hr post-superinfection (*Figure 2C*). Therefore, forskolin-triggered reactivation results in slower synthesis of Us11-GFP than superinfection. In addition, these data highlight the ability of forskolin to trigger reactivation from only a subpopulation of latently-infected neurons (approximately 1 in every 3.4 neurons compared to superinfection).

The production of infectious virus also mirrored the data for the detection of Us11-GFP-positive neurons, with a robust increase in viral titers between 24 hr and 60 hr post-stimulus (*Figure 2—figure supplement 1A*), which reflects both release of infectious virus from reactivating neurons and potentially cell-to-cell spread as WAY150138 could not be included. An increase in viral genome copy number was also not detected until 48 hr post-stimulus, which continued between 48 hr and 72 hr (*Figure 2—figure supplement 1B*). The quantification of viral genome copy number was also carried out in presence of WAY-150138, therefore indicating that DNA replication occurs in reactivating neurons and not as a consequence of cell-to-cell spread.

Given the observed 48 hr delay in viral DNA replication and production of infectious virus, we were interested to determine if there was a Phase I wave of lytic gene expression that occurred prior to viral DNA replication. We therefore carried out RT-qPCR to detect representative immediate-early (*ICP27* and *ICP4*), early (*ICP8* and *UL30*), and late (*UL48* and *gC*) transcripts between 5 hr and 20 hr-post addition of forskolin (*Figure 2D–F* and S1C-E). For all six transcripts, a significant up-regulation of mRNA occurred at 20 hr post-treatment, including the true late gene *gC*, whose expression would usually only be stimulated following viral genome replication in the context of de novo lytic replication. Therefore, this indicates that lytic gene expression is induced prior to viral DNA replication and that forskolin does trigger a Phase I wave of lytic gene expression. Notably, we did detect small but reproducible induction of *ICP27* mRNA at 5 hr post-stimulus, followed by a second induction at 20 hr (*Figure 2D*), indicating that there is likely differential regulation of some viral lytic transcripts

during Phase I reactivation induced by forskolin that is distinct from both NGF-deprivation and de novo lytic infection.

To determine whether JNK and DLK were required Phase I gene expression in response to forskolin, we investigated viral mRNA levels following forskolin-mediated reactivation in the presence of the JNK inhibitor SP600125. We found a significant reduction in *ICP27* (2.2-fold), *UL30* (3.3-fold) and *gC* (5.5-fold) mRNA levels at 20 hr post-stimulus in the presence of SP600125 (*Figure 2G–I*). For all genes tested, there was no significant increase in mRNAs in the JNK inhibitor treated neurons compared to mock. We observed comparable results following treatment with the DLK inhibitor GNE-3511, with a 2.3-, 3-, 8.8-fold decrease in *ICP27*, *UL30,* and *gC* mRNAs, respectively, compared to forskolin treatment alone, and no significant increase in mRNA levels compared to the unreactivated samples (*Figure 2G–I*). To further confirm that DLK is required for Phase I gene expression following forskolin treatment, we depleted DLK protein using two independent shRNAs via lentivirus mediated transduction of latently infected sympathetic neurons. Transduction with the DLK targeting shRNA vectors resulted in >90-fold reduction in DLK protein levels compared to the shRNA control transduced neurons (*Figure 2J*). We observed a significant reduction in HSV reactivation and I*CP27*, *UL30,* and gC mRNA levels at 20 hr post-forskolin treatment following transduction with either DLK shRNA lentivirus compared to the shRNA control transduced neurons (*Figure 2K–L* and *Figure 2— figure supplement 1F–G*).

It was possible that in addition to JNK, other signal transduction proteins were important in forskolin-mediated reactivation. A previous study found that DLK can be activated by PKA, which is known to be activated by cAMP (*Hao et al., 2016*). However, using well a characterized inhibitor of PKA (KT 5720), we were unable to find a role for PKA in Phase I reactivation (*Figure 2—figure supplement 2B*), although full reactivation was inhibited (*Figure 2—figure supplement 2A*). PKA has a number of downstream targets, including the transcription factor CREB, which is also involved in cellular gene expression changes in response to neuronal stimulation. Although addition of a CREB inhibitor (666-15), inhibited full reactivation (*Figure 2—figure supplement 2C*) it did not inhibit Phase I gene expression (*Figure 2—figure supplement 2D*). Because we did not detect a role for PKA we also investigated two additional proteins that can respond to increased levels of cAMP and mediate downstream signaling responses (see *Figure 1B*); EPAC1/Rapgef1 (inhibited by ESI09) and EPAC2/Rapgef2 (inhibited by SQ22,536). Downstream targets of EPAC1 1 and 2 include ERK and PKC respectively (*Huang and Gu, 2017*). However, inhibition of EPAC1 with ESI09 did not inhibit forskolin-mediated reactivation (*Figure 2—figure supplement 2E*). SQ22,536 is known to inhibit both adenylate cyclase and EPAC2 (*Emery et al., 2013*). Given that we had already found that SQ22,536 inhibited forskolin-mediated reactivation (*Figure 1E*), to directly test the inhibition of EPAC2 by SQ22,536 in a way that bypasses adenylate cyclase we investigated the effect reactivation induced by 8-Bromo-cAMP. Addition of SQ22,536 did not prevent reactivation triggered by the cAMP mimetic (*Figure 2—figure supplement 2F*). Taken together, these data suggest that forskolin induces a Phase I wave of gene expression that does not depend on activation of PKA, EPAC1 or EPAC2 but does require DLK and JNK activity. Because additional targets of cAMP in neurons include cyclic nucleotide-gated ion channels, we turned our attention to the role of hyperexcitability in HSV reactivation.

## Forskolin triggers a Phase I wave of viral gene expression that is independent of histone demethylase activity

Hyperexcitability results in the propensity of neurons to fire repeated action potentials, and is associated with specific changes in histone posttranslational modifications and accumulation of nuclear cFOS. This includes increased levels of γH2AX, a histone posttranslational modification linked to physiological DNA damage (*Alt and Schwer, 2018*; *Madabhushi et al., 2015*), which can be measured by the intensity of staining in neuronal nuclei. Forskolin treatment was associated with an increase in the levels of γH2AX at 5 hr post-treatment, which resolved by 15 hr post-treatment (*Figure 3—figure supplement 1A and C*), and is therefore indicative of physiological DNA damage and repair, which occurs upon neuronal hyperexcitability. To also indirectly probe for neuronal hyperexcitability following forskolin treatment, we also quantified nuclear cFOS accumulation and found that the intensity increased at 5 hr post-forskolin treatment (*Figure 3—figure supplement 1D*).

A second reason for probing the DNA damage/repair pathway in response to forskolin treatment is that an elegant study from the Huang lab found that reactivation of HSV from latency was

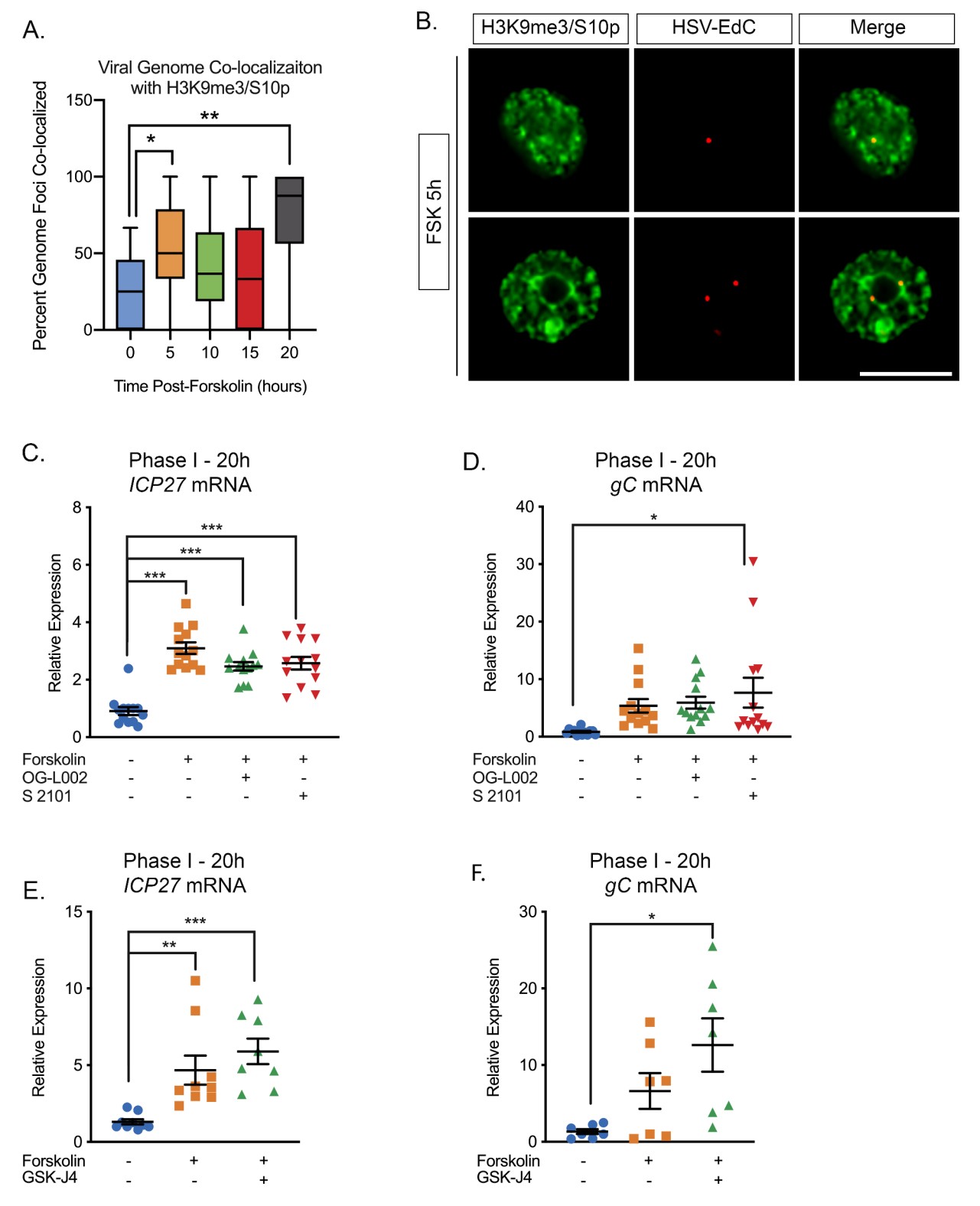

**Figure 3.** The Initial wave of viral lytic gene expression during forskolin-mediated reactivation is independent on histone demethylase activity. (**A**) Quantification of the percentage of genome foci stained using click-chemistry that co-localize with H3K9me3/S10p. At least 15 fields of view with 1–8 genomes per field of view were blindly scored from two independent experiments. Data are plotted around the median, with the boxes representing the 25th–75th percentiles and the whiskers the 1st-99th percentiles. (**B**) Representative images of click-chemistry based staining of HSV-EdC genomes and

*Figure 3 continued on next page*

*Figure 3 continued*

H3K9me3/S10p staining at 5 hr post-forskolin treatment. (C and D). Effect of the LSD1 inhibitors OG-L002 and S 2101 on forskolin-mediated Phase I of reactivation determined by RT-qPCR for *ICP27* (C) and *gC* (D) viral lytic transcripts at 20 hr post-forskolin treatment and in the presence of 15 μM OG-L002 and 20 μM S 2102. (E) Effect of the JMJD3 and UTX inhibitor GSK-J4 (2 μM) on forskolin-mediated Phase I measured by RT-qPCR for viral lytic transcripts ICP27 (E) and gC (F) at 20 hr post-forskolin treatment and in the presence of GSK-J4. For C-F each experimental replicate along with the mean and SEM is represented. (C–F). Statistical comparisons were made using a one-way ANOVA with a Tukey's multiple comparison. *p<0.05, **p<0.01, ***p<0.001.

The online version of this article includes the following source data and figure supplement(s) for figure 3:

**Source data 1.** Quantification of genome co-localization and RT-qPCR for *Figure 3*.
**Figure supplement 1.** Forskolin induces hyperexcitability-associated chromatin changes, and hsv reactivation that requires histone demethylase.
**Figure supplement 1—source data 1.** Quantification of nuclear staining intensity and GFP-positive neurons *Figure 3—figure supplement 1*.

associated with perturbation of the DNA damage/repair response (*Hu et al., 2019*). In this study, both inhibition of repair and exogenous DNA damage resulted in loss of AKT phosphorylation by PHLPP1, which was required for HSV reactivation. Although we did observe increased levels of γH2AX following forskolin treatment, we did not detect a concurrent loss of pAKT measured at 15 hr post-treatment (*Figure 3—figure supplement 1E*), whereas PI3-kinase inhibition by LY294002 did result in loss of pAKT. Both PI3-kinase inhibition and forskolin treatment did result in activation of the JNK cell stress pathway, indicated by increased c-Jun phosphorylation. This indicates that HSV reactivation and JNK activation in response to forskolin treatment does not involve dephosphoryla- tion of AKT. Therefore, hyperexcitability triggers reactivation via an alternative mechanism that does not feed into AKT phosphorylation.

Previously, we found that Phase I reactivation is accompanied with a JNK-dependent histone methyl/phospho (marked by H3K9me3/pS10) switch on lytic promoters (*Cliffe et al., 2015*). In corti- cal neurons, one study has found that hyperexcitability results in increased H3K9me3/pS10 (*Noh et al., 2015*). Therefore, we were particularly interested to determine whether forskolin treat- ment of sympathetic neurons triggered a histone S10 phosphorylation on H3K9me3. Forskolin trig- gered a transient increase in H3K9me3/S10p at 5 hr post-treatment that had returned to baseline by 10 hr (*Figure 3—figure supplement 1A and B*). This indicates that, in keeping with cortical neurons, forskolin induces a histone H3K9me3/pS10 methyl/phospho switch on regions on cellular chromatin.

We next sought to determine whether the phospho/methyl switch that arises as a result of hyper- excitability plays a role in Phase I of HSV reactivation. We therefore investigated whether viral genomes were co-localized with H3K9me3/S10p following forskolin treatment. To visualize HSV genomes, viral stocks were grown in the presence of EdC as described previously (*Alandijany et al., 2018*; *McFarlane et al., 2019*). Click-chemistry was performed on latently infected neurons follow- ing forskolin treatment. As shown in *Figure 3A and B*, viral genomes co-localized with H3K9me3/ pS10 following robust H3K9me3/S10p staining at 5 hr. The percentage of viral genomes that co- localized with H3K9me3/S10p was significantly increased compared to the mock reactivated samples at 5 hr and 20 hr post-forskolin treatment (*Figure 3A*).

Serine phosphorylation adjacent to a repressive lysine modification is thought to permit transcrip- tion without removal of the methyl group (*Gehani et al., 2010*; *Noh et al., 2015*). Therefore, we investigated whether histone demethylase activity was required for the initial induction in lytic gene expression following forskolin treatment. Previously, the H3K9me2 histone demethylase LSD1 has been found to be required for full HSV reactivation (*Liang et al., 2009*; *Hill et al., 2014*), and in our in vitro model this was determined by the synthesis of late viral protein at 48–72 hr post-reactivation (*Cliffe et al., 2015*). The addition of two independent LSD1 inhibitors (OG-L002 and S 2102) inhib- ited Us11-GFP synthesis at 72 hr post-reactivation (*Figure 3—figure supplement 1F*). Hence, LSD1 activity, and presumably removal of H3K9-methylation, is required for forskolin-mediated reactiva- tion. However, LSD1 inhibition did not prevent the initial induction of *ICP27* and *g*C mRNA expres- sion at 20 hr post-forskolin treatment (*Figure 3C and D*). Therefore, this initial wave of viral lytic gene expression following forskolin-mediated reactivation is independent of histone H3K9 demethy- lase activity.

We previously found that H3K27me demethylase activity is required for full reactivation but not the initial wave of gene expression (*Cliffe et al., 2015*). However, because of the lack of an antibody that specifically recognizes H3K27me3/S28p and not also H3K9me3/S10p (*Cliffe et al., 2015*), we

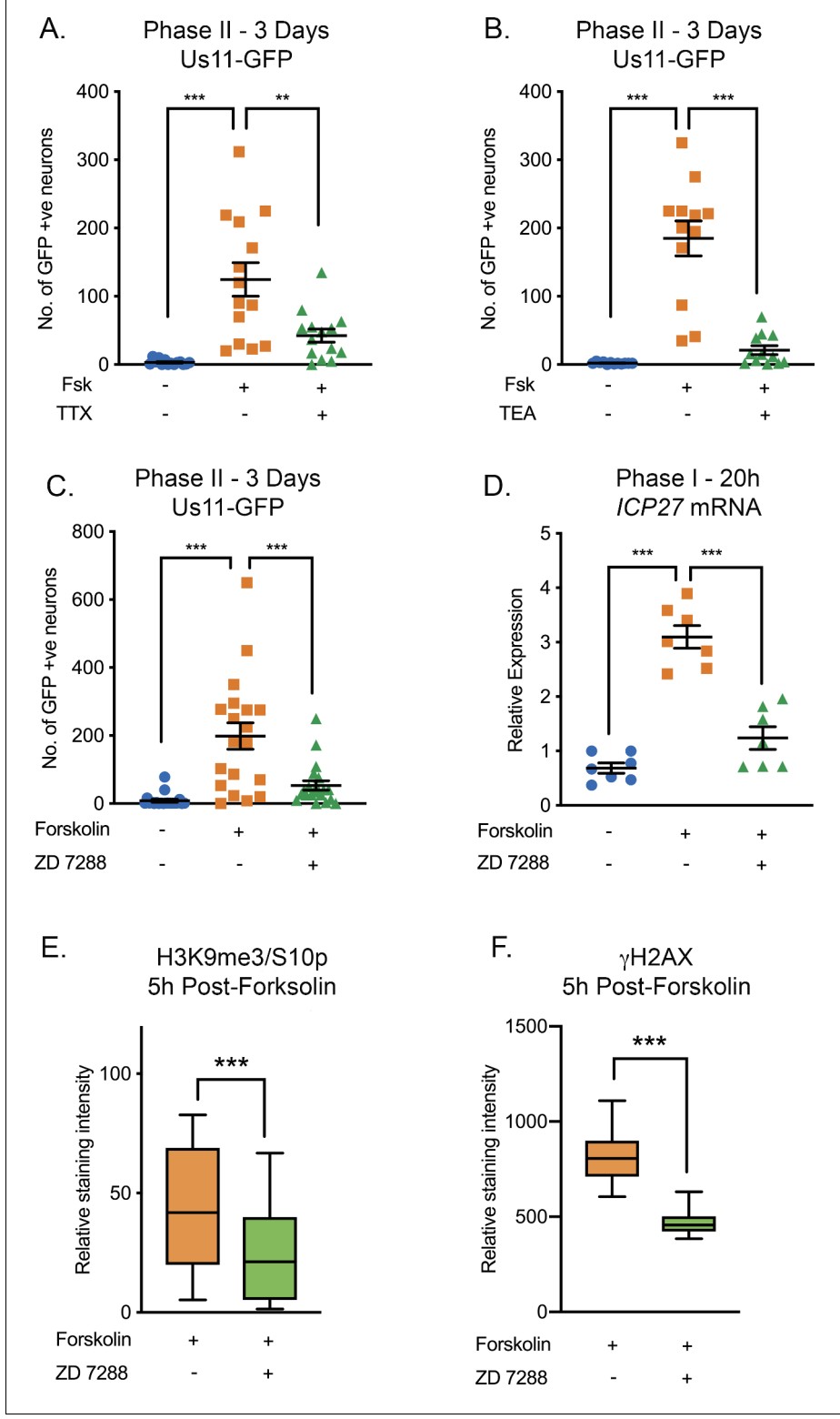

**Figure 4.** HSV Reactivation Mediated by Forskolin Requires Neuronal Excitability. (**A**) Latently infected cultures were reactivated with forskolin in the presence of the voltage-gated sodium channel blocker tetrodotoxin (TTX; 1 µM) and the number of Us11-GFP-positive neurons quantified at 3 days post-reactivation. (**B**) Latently infected cultures were reactivated with forskolin in the presence of the voltage-gated potassium channel blocker tetraethylammonium (TEA; 10 mM) and the number of Us11-GFP-positive neurons quantified at 3 days post-

*Figure 4 continued on next page*

*Figure 4 continued*

reactivation. (**C**) Forskolin-mediated reactivation in the presence of the HCN channel blockers ZD 7288 (10μM) quantified as the numbers of Us11-GFP-positive neurons at 3 days post-reactivation. (**D**) The effect of ZD 7288 on the HSV lytic gene transcript ICP27 during Phase I reactivation measured at 20 hr post-forskolin treatment by RT-qPCR. In A-D individual experimental replicates are represented along with the mean and SEM. (**E and F**) Quantification of the relative nuclear staining for H3K9me3/S10p and γH2AX in SCG neurons at 5 hr post-forskolin treatment and in the presence of ZD 7288 from >800 cells/condition from two independent experiments. Data are plotted around the mean, with the boxes representing the 25th-75th percentiles and the whiskers the 5st-95th percentiles. Statistical comparisons were made using a one-way ANOVA with a Tukey's multiple comparison (**A–D**) or two-tailed unpaired t-test (**E–F**). *p<0.05, **p<0.01, ***p<0.001. In A-D individual experimental replicates are represented.

The online version of this article includes the following source data and figure supplement(s) for figure 4:

**Source data 1.** Quantification of GFP-positive neurons, RT-qPCR and nuclear staining intensity for *Figure 4*.
**Figure supplement 1.** HSV Reactivation Mediated by Forskolin Requires Neuronal Excitability.
**Figure supplement 1—source data 1.** Quantification of GFP-positive neurons and RT-qPCR for *Figure 4—figure supplement 1*.

are unable at this point to investigate genome co-localization with this combination of modifications. However, we could investigate the role of the H3K27me demethylases in forskolin-mediated reactivation. Treatment of neurons with the UTX/JMJD3 inhibitor GSK-J4 (*Kruidenier et al., 2012*) prevented the synthesis of Us11-GFP at 72-hr post-reactivation, indicating that removal of K27 methylation is required full reactivation (*Figure 3—figure supplement 1G*). However, the initial burst of gene expression (assessed by *ICP27* and *gC* mRNA levels) was robustly induced at 20 hr post-forskolin treatment in the presence of GSK-J4 (*Figure 3E and F*). Taken together, our data indicate that the initial phase of gene expression following forskolin treatment is independent of histone demethylase activity and therefore consistent with a role for a histone methyl/phospho switch in permitting lytic gene expression.

## Forskolin-mediated reactivation requires neuronal excitability

Given that the HSV genome co-localized with regions of hyperexcitability-induced changes in histone phosphorylation, we investigated whether reactivation was linked to neuronal excitability. To inhibit action potential firing, we treated neurons with tetrodotoxin (TTX), which inhibits the majority of the voltage-gated sodium channels and consequently depolarization. The addition of TTX significantly inhibited HSV reactivation triggered by forskolin, as measured by Us11-GFP-positive neurons at 72-hr post-stimulus (*Figure 4A*). To further confirm a role for repeated action potential firing in forskolin-mediated reactivation, we investigated the role of voltage-gated potassium channels, which are required for membrane repolarization. The addition of tetraethylammonium (TEA), which inhibits voltage-gated potassium channel activity, also blocked HSV reactivation measured by Us11-GFP-positive neurons at 3 days post-forskolin treatment (*Figure 4B*). Taken together, these data indicate that action potential firing is required for forskolin-mediated reactivation.

Increased levels of cAMP can act on nucleotide-gated ion channels, including the hyperpolarization-activated cyclic nucleotide-gated (HCN) channels. HCN channels are K+ and Na+ channels that are activated by membrane hyperpolarization (*Sartiani et al., 2017*; *Kullmann et al., 2016*). In the presence of high levels of cAMP, the gating potential of HCN channels is shifted in the positive direction, such that HCN channels can open at resting membrane potential, resulting in an increased propensity of neurons to undergo repeated firing (*Kullmann et al., 2016*; *DiFrancesco and Tortora, 1991*; *Kase and Imoto, 2012*). HCN channel activity inhibitors include ZD 7288, Ivabradine, or cesium chloride. ZD 7288 has been characterized as an open-state blocker of HCN channels, however there is also evidence that it can inhibit voltage-gated sodium channel activity (*Wu et al., 2012*). This combined effect of ZD 7288 is a plus as it operates via multiple mechanism to inhibit neuronal excitability. Ivabradine is an FDA approved HCN inhibitor that has been demonstrated to specifically inhibit all four HCN channels (*Novella Romanelli et al., 2016*). Cesium chloride is a non-selective cation channel blocker. Addition of ZD 7288 (*Figure 4C*), Ivabradine (*Figure 4—figure supplement 1A*) or CsCl (*Figure 4—figure supplement 1B*) all significantly reduced HSV reactivation triggered by forskolin, as measured by Us-11 GFP-positive neurons at 3 days post-stimulus. To

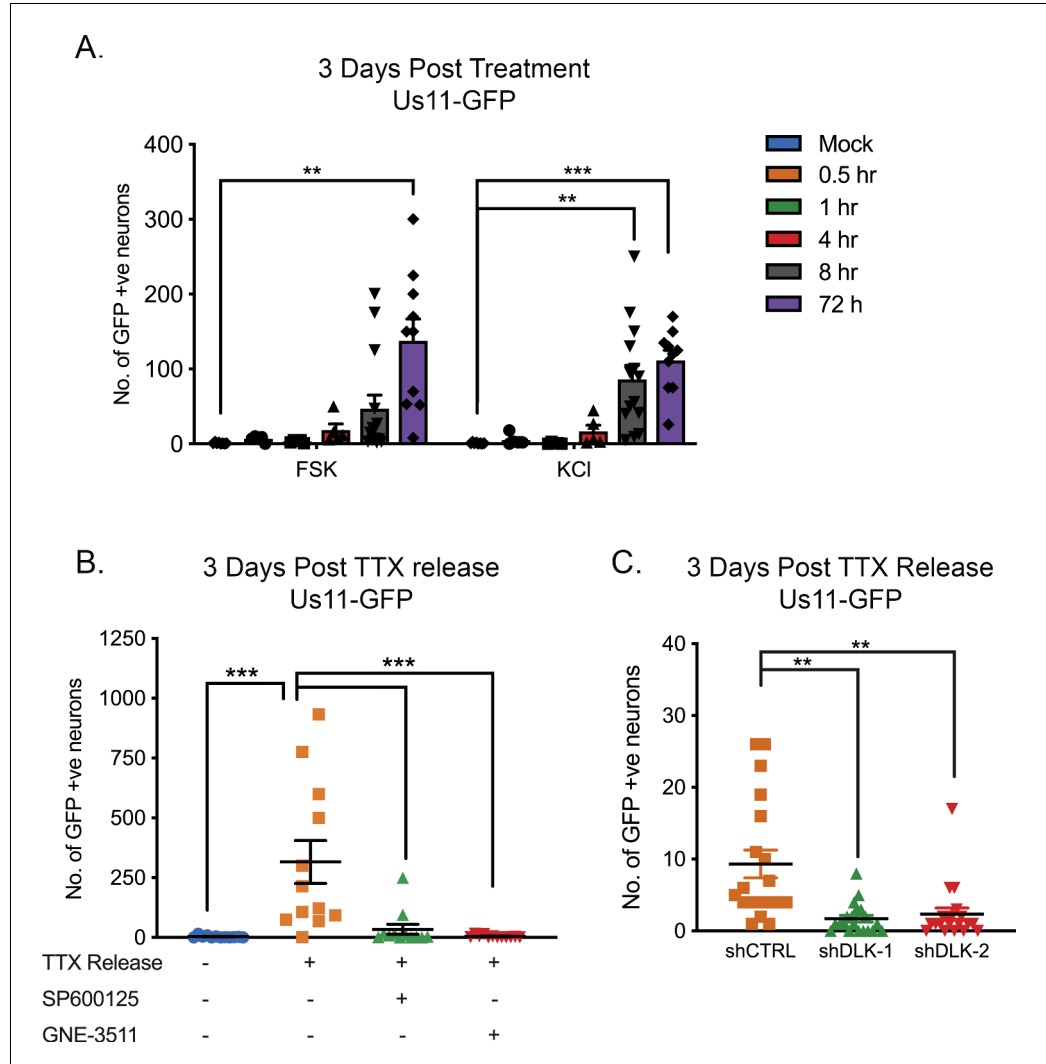

**Figure 5.** HSV Reactivation triggered by prolonged neuronal hyperexcitability is DLK/JNK-dependent. (**A**) Latently infected SCG cultures were treated with forskolin or KCl (55 mM) for the indicated times followed by wash-out. Reactivation was quantified by number of Us11-GFP-positive neurons at 3 days after the initial stimulus was added. (**B**) Latently infected neurons were placed in tetrodotoxin (TTX; 1 μM) for 2 days and the TTX was then washed out. At the time of wash-out the JNK inhibitor SP600125 (20 μM) or DLK inhibitor GNE-3511 (4 μM) was added. (**C**) Latently infected neurons were transduced with either control non-targeting shRNA or shRNA targeting DLK for 3 days, then placed in tetrodotoxin (TTX; 1 μM) for 2 days and the TTX was then washed out. Reactivation was quantified at 3 days post-wash-out. Individual experimental replicates, the mean and SEMs are represented. Statistical comparisons were made using a one-way ANOVA with a Tukey's multiple comparison. **p<0.01, ***p<0.001.

The online version of this article includes the following source data for figure 5:

**Source data 1.** Quantification of GFP-positive neurons for *Figure 5*.

determine the contribution of HCN channel activity and neuronal excitability to the initial induction of HSV lytic mRNA expression, we assessed viral mRNA expression during Phase I in the presence and absence of ZD 7288. Expression of representative lytic mRNAs *ICP27* (*Figure 4D*), *UL30* and *gC* (*Figure 4—figure supplement 1C and D*) were significantly decreased in the presence of ZD 7288 compared to the forskolin treated neurons alone, and were not significantly increased compared to the mock treated samples.

We also confirmed that neuronal excitation was required for the global changes in histone phosphorylation observed with exposure of sympathetic neurons to forskolin. Addition of ZD 7288

resulted in significantly decreased staining intensities of both H3K9me3/S10p and γH2AX at 5 hr post-forskolin treatment (*Figure 4E and F*), which was the peak time-point for which we observed these changes upon forskolin treatment alone (*Figure 3—figure supplement 1B* and S3C). Therefore, activity of the HCN channels and/or voltage-gated sodium channels in response to increased levels of cAMP, results in hyperexcitability-associated changes in histone modifications and the initial induction of lytic gene expression during Phase I and reactivation of HSV from latent infection.

## HSV reactivation can be induced by stimuli that directly increase neuronal excitability

The role of ion channel activity in forskolin-mediated reactivation prompted us to investigate whether additional stimuli that induce hyperexcitability in neurons also trigger HSV reactivation. We were also interested in whether reactivation required chronic versus short-term hyperexcitability. Increasing the extracellular concentration of KCl is well-known to induce action potential firing. Therefore, we investigated the timing of both KCl and forskolin-mediated hyperexcitability in HSV reactivation. Both of these treatments triggered HSV reactivation more robustly if applied for 8 hr or more (*Figure 5A*). This indicates that chronic neuronal hyperexcitability is important in inducing reactivation of HSV.

To further clarify that hyperexcitability can directly trigger HSV reactivation, we investigated the effects of removal from a TTX block on latently infected neurons. The addition of TTX to neurons results in synaptic scaling whereby a neuron's excitatory synaptic strength increases in response to inhibition of firing, so that when the TTX is removed the neurons enter a hyperexcitable state (*Ibata et al., 2008*; *Turrigiano et al., 1998*; *Lee and Kirkwood, 2019*; *Sokolova and Mody, 2008*). TTX was added to the neurons for 2 days and then washed out. This resulted in a robust HSV reactivation as determined by Us11-GFP synthesis (*Figure 5B*). We also investigated whether the JNK cell stress pathway was important in HSV reactivation in response to TTX-release. Addition of the JNK inhibitor SP600125 or the DLK inhibitor GNE-3511 blocked HSV reactivation following TTX-release (*Figure 5B*) as did shRNA mediated depletion of DLK (*Figure 5C*). Therefore, directly inducing neuronal hyperexcitability triggers HSV reactivation in a DLK/JNK-dependent manner.

## IL-1β triggers HSV reactivation in mature neurons in a DLK and voltage-gated sodium channel-dependent manner

Our data thus far point to reactivation of HSV following increasing episodes of neuronal hyperexcitability in a way that requires activation of the JNK cell stress pathway. However, we wished to link this response to a physiological trigger that may stimulate HSV reactivation in vivo. Increased HCN channel activity has been associated with inflammatory pain resulting from the activity of pyrogenic cytokines on neurons (*Emery et al., 2011*). In addition, IL-1β is known to act on certain neurons to induce neuronal excitation (*Vezzani and Viviani, 2015*; *Schneider et al., 1998*; *Binshtok et al., 2008*). IL-1β is released in the body during times of chronic, psychological stress. In addition, IL-1β contributes to the fever response (*Ericsson et al., 1994*; *Goshen and Yirmiya, 2009*; *Koo and Duman, 2009*; *Saper and Breder, 1994*). In sympathetic neurons, we found that exposure of mature neurons to IL-1β induced an accumulation of the hyperexcitability-associated histone post-translational modifications γH2AX and H3K9me3/S10p (*Figure 6A*, *Figure 6—figure supplement 1A–B*). We did not observe the same changes for post-natal neurons. The precise reasons for this maturation-dependent phenotype are unknown at this point but we hypothesize it could be due to changes in the expression of cellular factors required to respond to IL-1β. Therefore, these experiments were carried out on neurons that were postnatal day 36. The kinetics of induction of these histone modifications were different from what we had previously observed for forskolin treatment, as both γH2AX and H3K9me3/S10p steadily accumulated to 20 hr post-treatment. This likely reflects the activation of upstream signaling pathways in response to IL-1β prior to inducing neuronal excitation as IL-1β is known to increase the expression of voltage-gated sodium channels (*Binshtok et al., 2008*). To confirm that the increase in γH2AX and H3K9me3/S10p was linked to neuronal excitation we measured the staining intensity when IL-1β was added in the presence of TTX. We also added an IL1R neutralizing antibody to verify that the response was specific to signaling via IL-1 mediated binding to its receptor. Addition of either TTX or anti-IL1R antibody resulted in a significant

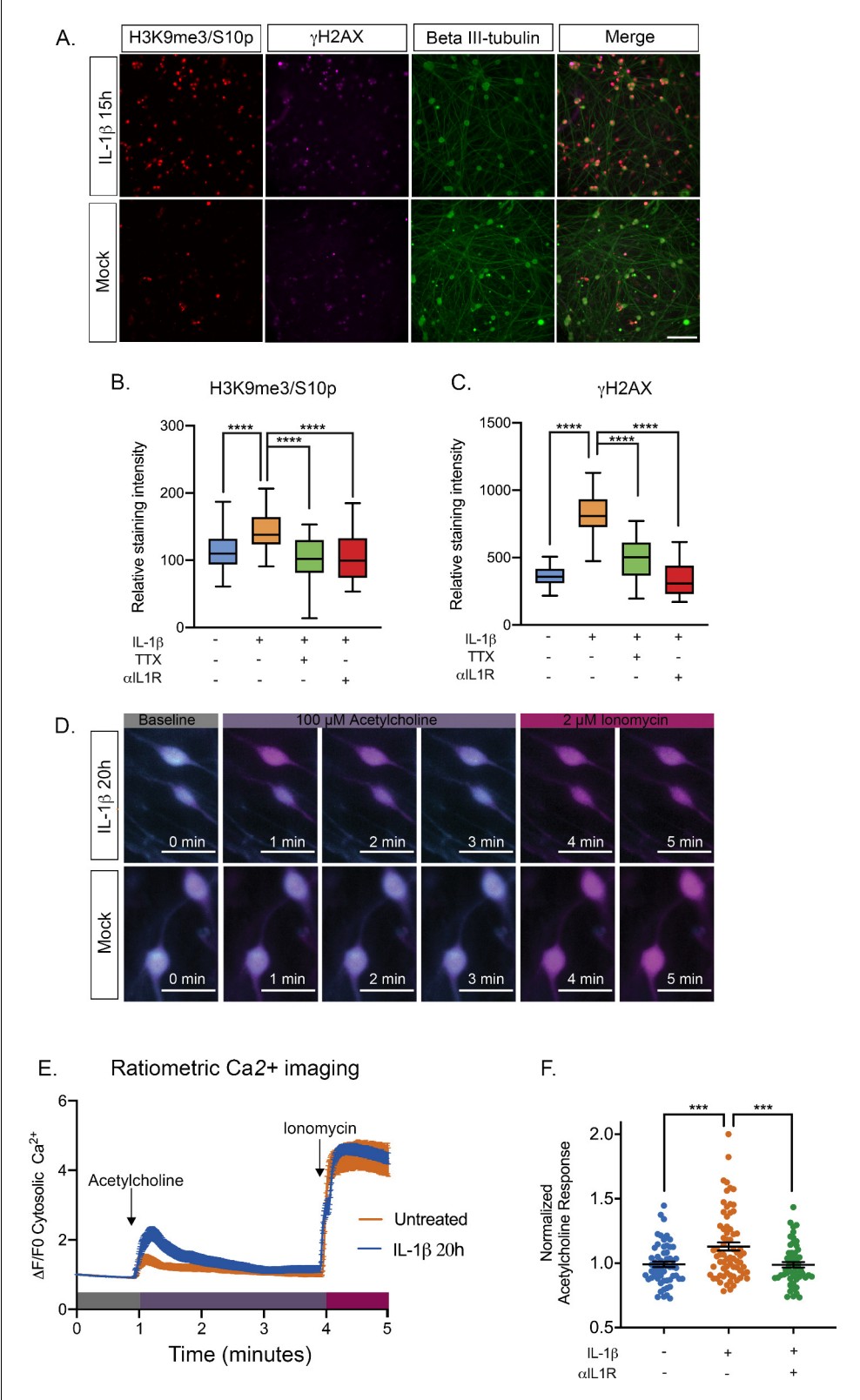

**Figure 6.** IL-1β Treatment of sympathetic neurons results in changes consistent with heightened neuronal excitability. (**A**) Adult P36 SCG neurons were treated with IL-1β (30 ng/mL) for 15 hr and stained for H3K9me3/S10p, γH2AX and beta II-tubulin to mark neurons. (**B and C**) Quantification of the intensity of H3K9me3/S10p (**B**) and γH2AX (**C**) staining following 15 of IL-1β treatment and in the presence of tetrodotoxin (TTX; 1 μM) or anti-IL1 receptor (IL-1R) blocking antibody (2 μg/mL). Data are plotted around the median and whiskers represent the 5th-95th percentiles. (**D**) Representative

*Figure 6 continued on next page*

*Figure 6 continued*

images of cytosolic Ca$^{2+}$ elevations measured using Fura-2-AM in neurons stimulated with 100 μM acetylcholine either pre-treated with IL-1β for 20 hr or mock treated. As a control the neurons were also treated with Ionomycin at the end of the protocol. Bar = 100 μm. (E) Representative experiment for cytosolic Ca$^{2+}$ elevations in neurons stimulated with 100 μM acetylcholine. Cells were pretreated with IL-1β or vehicle for 20 hr prior to imaging. The plotted values were calculated as a change in fluorescence/initial fluorescence (ΔF/F0). Error bars represent SEM (IL-1β treatment, n = 58 cells and vehicle control, n = 25 cells). (F) Peak cytosolic Ca$^{2+}$ elevations normalized to untreated controls in neurons stimulated with 100 μM acetylcholine. Cells were pretreated with IL-1β (n = 70, wells) or vehicle (n = 58, wells) for 20 hr prior to imaging. IL-1R blocking antibody (n = 54, wells) was also added. Data points represent individual wells, horizontal line represents mean. Statistical comparisons were made using a one-way ANOVA with a Tukey's multiple comparison (B–D). ***p<0.001 ****p<0.0001.

The online version of this article includes the following source data and figure supplement(s) for figure 6:

**Source data 1.** Quantification of nuclear staining intensity and ratiometric calcium imaging for *Figure 6*.

**Figure supplement 1.** IL-1β Treatment of mature scg neurons induces excitability-associated histone post-translational modifications (supplement to *Figure 6*).

**Figure supplement 1—source data 1.** Quantification of nuclear staining intensity for *Figure 6—figure supplement 1*.

reduction in γH2AX and H3K9me3/S10p levels in response to IL-1β treatment (*Figure 6B–C*), indicating that the response is directly due to IL1 and requires activity of voltage-gated sodium channels.

To test whether IL-1β induces a hyperexcitable state in sympathetic neurons, we measured cytosolic Ca$^{2+}$ elevations within neurons following addition of their cognate neurotransmitter, acetylcholine. Compared to mock treated controls, neurons that were pre-treated with IL-1β for 20 hr displayed higher elevations in cytosolic Ca$^{2+}$ as measured by Fura-2-AM, a ratiometric indicator of cytosolic Ca$^{2+}$ (*Figure 6D–E*). To control for any intrinsic artifacts in dye loading or retention, neurons were also treated with Ca$^{2+}$ ionophore, ionomycin, to raise the intracellular levels of Ca$^{2+}$ directly at the end of the recording protocol. We observed nearly identical elevations in ionomycin-mobilized Ca$^{2+}$ in both untreated and IL-1β treated neurons. Importantly, compared to untreated controls, neurons pre-treated with IL-1β exhibit significantly higher levels of cytosolic Ca$^{2+}$ in response to acetylcholine. When IL-1β pre-treated sympathetic neurons were measured on a population basis for their response to acetylcholine (*Figure 6E*), we did observe a range of responses, which likely reflects that these are a heterogenous population of mature neurons that vary in response to IL-1β as well as acetylcholine. Importantly, we did detect a significant increase in intracellular Ca$^{2+}$ in response to acetylcholine in neurons that we treated with IL-1β. The increase in acetylcholine responses observed in IL-1β treated neurons was prevented with the addition of the IL1R blocking antibody, indicating that it is specific for signaling through the IL1 receptor (*Figure 6F*).

Because IL-1β was found to cause sympathetic neurons to enter a hyperexcitable state, we went on to investigate whether IL-1β was able to induce HSV reactivation. Addition of IL-1β triggered HSV reactivation in mature neurons quantified by the number of Us11-GFP neurons at 3 days (*Figure 7A*). We did observe a large range in the numbers of reactivating neurons, which likely reflects the heterogeneity in responses to response to IL-1β. In addition, it is likely there is heterogeneity between different populations of latent viral genomes in terms of their chromatin structure and subnuclear localization, which also impacts their reactivation ability upon addition of a given stimulus. Addition of the IL-1R blocking antibody preventing IL-1β induced reactivation (*Figure 7B*). In a number of experiments, we did observe lower levels of reactivation than what we had previously observed upon forskolin treatment of younger neurons. Therefore, we also treated mature neurons with forskolin and saw similar levels of reactivation than those observed with IL-1β (*Figure 7B*), indicating that latently infected mature neurons may be more restricted for reactivation.

Inhibition of voltage-gated sodium channels by TTX resulted in a significant decrease in the ability of IL-1β to induce reactivation (*Figure 7E*), therefore indicating that IL-1β triggered reactivation is via increasing neuronal activity. Reactivation was reduced in the presence of the HCN-channel inhibitor ZD 7288, although this decrease was not significant (p=0.2255), perhaps suggesting that IL-1β induction of neuronal activity is not directly due to the action of cAMP on HCN channels and instead HCN channel activity may be required for maximal hyperexcitability and reactivation. Importantly, the addition of the DLK inhibitor GNE-3511 blocked reactivation in response to IL-1β (*Figure 7C*) and the role for DLK was confirmed by shRNA mediated depletion (*Figure 7D*). Therefore, IL-1β can induce sympathetic neurons to become hyperexcitable and trigger HSV-1 reactivation via activation of DLK.

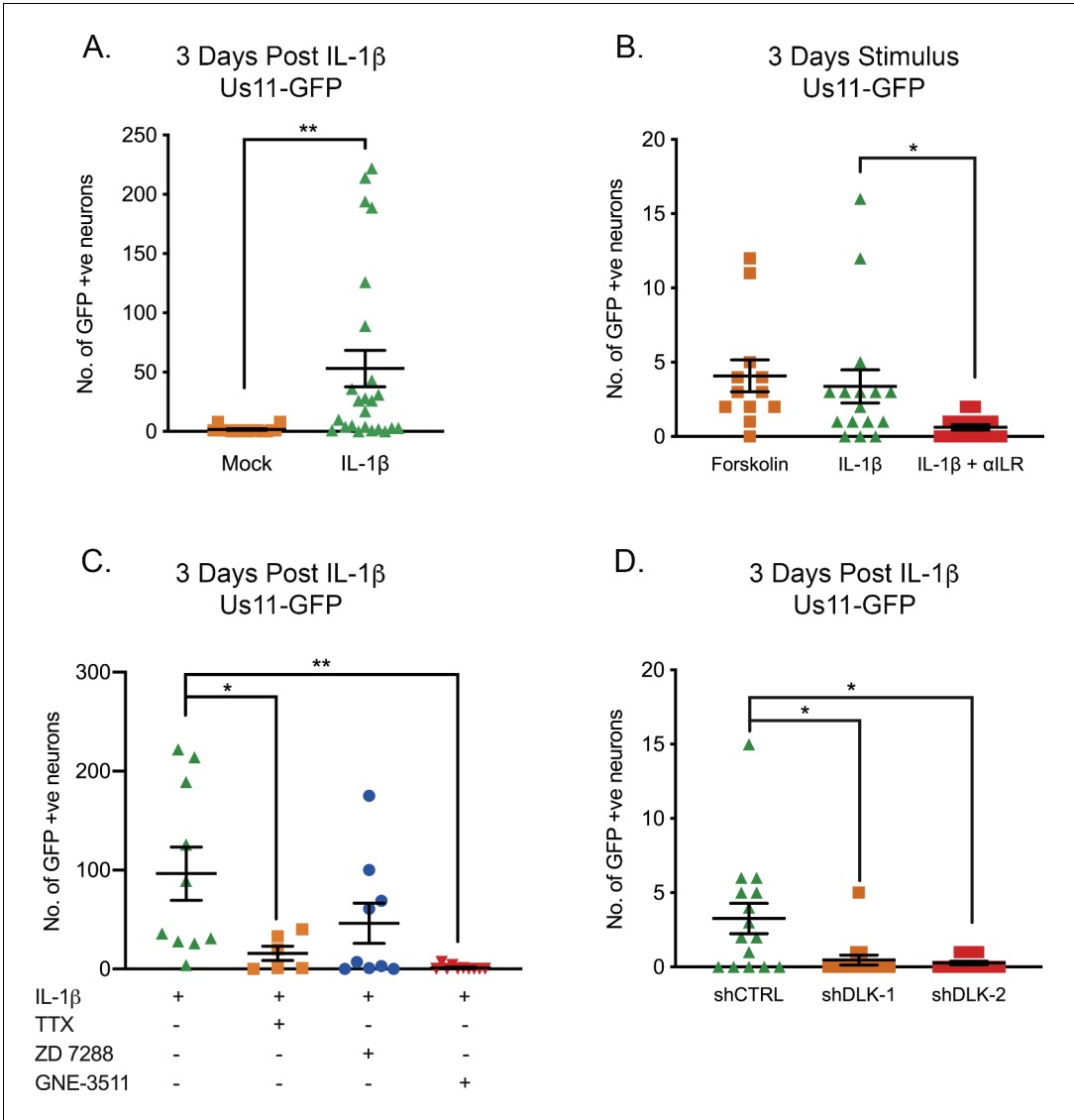

**Figure 7.** IL-1β-Induced HSV Reactivation is linked to heightened neuronal excitability and dlk activation. (**A**) Quantification of Us11-GFP expressing neurons following addition of IL-1β to latently infected cultures of mature SCG neurons. (**B**) Numbers of Us11-GFP-positive neurons following addition of forskolin or IL-1β to mature SCG neurons, and in the presence of an IL-1R-blocking antibody (2 μg/mL). (**C**) Quantification of IL-1β induced reactivation in the presence of the voltage-gated sodium channel blocker TTX (1 μM), the HCN-channel blocker ZD 7288 (10 μM) and the DLK inhibitor GNE-3511 (4 μM). (**D**) Latently infected SCG neurons were transduced with an shRNA control lentivirus or lentiviruses expressing shRNA against DLK. Three days later IL-1β was added to the cultures and the numbers of GFP-positive neurons quantified at 3 days later. Individual experimental replicates, means and SEMs are represented. Statistical comparisons were made using two-tailed unpaired t-test (**A**) or a one-way ANOVA with a Tukey's multiple comparison (**B–D**). *p<0.05, **p<0.01.

The online version of this article includes the following source data for figure 7:

**Source data 1.** Quantification of GFP-positive neurons for *Figure 7*.

## Discussion

As herpesviruses hide in the form of a latent infection of specific cell types, they sense changes to the infected cell, resulting in the expression of viral lytic genes and ultimately reactivation. HSV establishes latency in neurons and has previously been found to respond to activation of a neuronal stress signaling pathway (*Cliffe et al., 2015*). As an excitable cell type, the function of neurons is to rapidly transmit stimuli via the firing of action potentials, and under conditions of hyperexcitability,

neurons increase their propensity to fire repeated action potentials. Here we show that this state of hyperexcitability induces HSV to undergo reactivation in a DLK/JNK-dependent manner, indicating that the virus responds to both activation of cell stress signaling and prolonged hyperexcitability via a common pathway to result in reactivation. This common pathway also permits viral lytic gene expression from silenced promoters without the requirement of histone demethylase activity via a histone methyl/phospho switch. Conditions that result in hyperexcitability include prolonged periods of stress, sun burn and inflammation, which are both linked to the release of IL-1β (*Ericsson et al., 1994*; *Goshen and Yirmiya, 2009*; *Koo and Duman, 2009*; *Saper and Breder, 1994*; *Kupper et al., 1987*). Consistent with this, here we show that IL-1β induces DNA damage and histone H3 phosphorylation in sympathetic neurons, which are both markers of neuronal excitability, and causes increased responses to neurotransmitter stimulation. Importantly, IL-1β triggered HSV reactivation that was dependent on neuronal activity and activation of DLK. Therefore, this study identifies a physiological stimulus that induces HSV reactivation via increasing neuronal excitability and places DLK/JNK signaling and a histone phospho/methyl switch as central to HSV reactivation.

In this study, we employed an in vitro system using primary murine sympathetic neurons. In vitro systems using primary neurons from rats and mice or differentiated embryonic human neurons are now being used in many laboratories to elucidate molecular mechanisms of HSV latency and reactivation (*Thellman and Triezenberg, 2017*; *Edwards and Bloom, 2019*; *D'Aiuto et al., 2019*). Model systems for lytic replication commonly use fetal or new born fibroblasts. These model systems come with the caveat that viral replication, entry into latency and reactivation take place in the absence of the host immune response. In particular for latency, where there is a complex interplay between the host immune response and infected neurons during the establishment and reactivation from latency, and latency in humans occurs over a period of decades. There is also evidence of persistent immune infiltration into the ganglia that correlates with herpesvirus latency (*Theil et al., 2003*). However, in vitro systems are powerful for identifying neuronal pathways important for different phases of reactivation reactivation, adding back components of the host immune response at specific times and investigating the role of viral gene products solely in reactivation.

Experiments using primary neuronal in vitro model systems and inducing reactivation by PI3-kinase inhibition have shown that reactivation in these models involves two phases. Importantly, these in vitro models permit the dissection of rapid molecular events that may be difficult to observe in in vivo models. Phase I involves the synchronous up-regulation of lytic gene expression that occurs independently of the viral transactivator VP16 and the activity of cellular histone demethylases (*Kim et al., 2012*; *Cliffe et al., 2015*). Synchronous induction of lytic gene expression has also been observed in an ex vivo model of HSV reactivation induced by axotomy combined with inhibition of NGF-signaling (*Du et al., 2011*). A population of neurons progresses to full reactivation (Phase II), which is dependent on both VP16 and HDM activity (*Kim et al., 2012*; *Cliffe et al., 2015*). We previously found that lytic gene expression in Phase I is DLK/JNK-dependent and is correlated with a JNK-dependent histone methyl/phospho switch on lytic gene promoters (*Cliffe et al., 2015*). Here we demonstrate that a Phase I wave of viral gene expression that is dependent on activation of JNK but not histone demethylases also occurs in response to forskolin. The co-localization of viral genomes with H3K9me3/pS10 indicates that a histone methyl/phospho switch also permits lytic gene expression to occur following forskolin treatment in a manner that is independent of HDM activity. This indicates that reactivation proceeds via a Phase I-wave of gene expression in response to multiple different stimuli. However, we note that there may be differences in the mechanism and kinetics of reactivation with different stimuli and/or strains of HSV-1 as reactivation triggered by axotomy or heat shock following infection with a more pathogenic strain of HSV may bypass Phase I or occur more rapidly, making Phase I difficult to detect (*Cliffe and Wilson, 2017*; *Liang et al., 2009*; *Doll et al., 2020*). It will be especially interesting to determine in the future whether there are differences in the progression to reactivation with different strains of HSV, further elucidate the underlying progression to reactivation and requirements for Phase I. Ultimately, reactivation kinetics may relate to differences in the epigenetic structures of viral genomes that vary based on virus strains or differential manipulation of host-cell signaling pathways.

The host immune response should also be taken into account when considering the progression to full reactivation. The interplay between the host immune response and progression to reactivation is complex and differential responses likely inhibit, or as this study suggests, even promote different phases of the reactivation pathway. The presence of IFN can prevent Phase I reactivation

(*Linderman et al., 2017*). CD8 T cells have been linked to controlling HSV reactivation (*Divito et al., 2006*), and regulatory T cells have been found to facilitate HSV reactivation in an in vivo model by suppressing CD8 T cells (*Yu et al., 2018*). However, a recent in vivo study found that Iba+ phagocytic cells play a key role clearing reactivating neurons (*Doll et al., 2020*), which required viral DNA replication. HSV latency is not fully silent, with detectable lytic gene expression in infected neurons (*Ma et al., 2014*; *Singh and Tscharke, 2020*). Stimuli, including IL-1, enhanced neuronal excitation and neuronal stress likely contribute to bursts of lytic gene expression in vivo, but full reactivation could be controlled by the host immune response. Because of the potential link between HSV infection and the progression of neurodegenerative disease, understanding the mechanism of 'leaky' latency, how this is controlled by the host immune response and any potential effects on neuronal function will be especially important.

The Wilcox lab demonstrated in 1992 that reactivation can be induced by forskolin, and it has since been used as a trigger in multiple studies (*Smith et al., 1992*; *Colgin et al., 2001*; *De Regge et al., 2010*; *Danaher et al., 2003*). However, the mechanism by which increasing levels of cAMP induces lytic gene expression was not known. Here, we link cAMP-induced reactivation to the excitation state of the neuron and show that the initial induction of viral gene expression is dependent on DLK and JNK activity but independent of CREB and PKA. The activity of PKA may be required for full reactivation, which is also consistent with a role for PKA in overcoming repression of the related Pseudorabies Virus during de novo axonal infection (*Koyuncu et al., 2017*). Our data also suggest that CREB may be involved in the progression to full reactivation. However, the mechanism of action of the inhibitor used here, 666–15, is not entirely clear. It has been reported as preventing CREB-mediated gene expression, but may act to prevent recruitment of histone acetyltransferases (*Xie et al., 2015*). Therefore, inhibition of Phase II reactivation by 666–15 would be consistent with more large-scale chromatin remodeling on the viral genome at this stage. In addition, previous work has identified a role for inducible cAMP early repressor (ICER) in HSV reactivation (*Colgin et al., 2001*). ICER is a repressor of gene expression that acts via heterodimerization with members of the CREB/ATF family of transcription factors. CREB expression is also down-regulated by loss of NGF-signaling (*Riccio et al., 1999*), a known trigger of HSV reactivation. Therefore, it is conceivable that inhibition, rather than activation, of CREB is important for reactivation of HSV from latency.

Neuronal hyperexcitability results in DNA damage followed by repair, which together are thought to mediate the expression of cellular immediate-early genes (*Alt and Schwer, 2018*; *Madabhushi et al., 2015*). Here we show that forskolin treatment and IL-1β also induce DNA damage in sympathetic neurons. Previously, HSV reactivation has been found to occur following inhibition of DNA damage, inhibition of repair, and exogenous DNA damage (*Hu et al., 2019*). In the context of repair inhibition or exogenous DNA damage, reactivation was dependent on dephosphorylation of AKT by the PHLPP1 phosphatase and activation of JNK, and therefore feeds into the same pathway as PI3K-inhibition. However, we did not observe decreased AKT phosphorylation in response to forskolin treatment, indicating that the mechanism of reactivation is distinct following physiological levels of DNA damage resulting from neuronal hyperexcitability versus perturbation of the damage/repair pathways.

Conditions that result in hyperexcitability include prolonged periods of stress and inflammation, which are both linked to the release of IL-1β (*Ericsson et al., 1994*; *Goshen and Yirmiya, 2009*; *Koo and Duman, 2009*; *Saper and Breder, 1994*). Consistent with these findings, we show that IL-1β treatment induces two markers of neuronal excitability, DNA damage and histone H3 phosphorylation, in primary sympathetic neurons in addition to promoting a heightened excitation response to acetylcholine. The IL-1 family of cytokines act via the IL-1 receptor to activate downstream signaling pathways (*Weber et al., 2010*). IL-1α, which also signals via the IL-1R, is released locally as an alarmin. Interestingly, IL-1α and IL-1β are found at high levels in keratinocytes and are released upon HSV-1 infection (*Orzalli et al., 2018*), where they can mediate antiviral responses in underlying stromal fibroblasts and endothelial cells. Notably, upon UVB radiation exposure, keratinocytes and corneal epithelial cells upregulate and release IL-1α and IL-1β (*Kupper et al., 1987*; *Keadle et al., 2000*), potentially linking cytokine and alarmin release from keratinocytes to reactivation of HSV following UV damage. It should also be noted that additional cytokines and growth factors are released from keratinocytes upon damage, including NGF (*Tron et al., 1990*). Although deprivation of NGF results in activation of a neuronal cell stress pathway that can induce HSV reactivation (*Cliffe et al., 2015*), exposure of neurons to increased levels of NGF can cause them to become hyperexcitable

(*McMahon, 1996*). Therefore, it is possible that the correct balance of neurotrophin levels may balance HSV latency, akin to how the correct balance of DNA damage and repair is also required to maintain latency (*Hu et al., 2019*; *Cliffe, 2019*). It will be incredibly interesting to determine how neurotrophin balance, cytokine release and hyperexcitability converge to regulate HSV latency in vivo. Although there is no direct evidence as yet for neuronal excitability inducing reactivation in an in vivo model, a commonly used trigger for HSV reactivation, axotomy, can directly induce hyperexcitability and also result in IL-1β release from satellite glial cells (*Hanani and Spray, 2020*). Thermal stress is used as an in vivo trigger of HSV reactivation (*Sawtell and Thompson, 1992*), which can also cause increased neuronal firing of nociceptor sensory neurons (*Paricio-Montesinos et al., 2020*). IL-6, which is also a known-inducer of neuronal hyperexcitability (*Vezzani and Viviani, 2015*), has been linked to heat stress-induced HSV reactivation (*Noisakran et al., 1998*). However, a direct role for hyperexcitability and/or IL-1 remains to be explored in this or other in vivo models of HSV reactivation.

Previously, we found that JNK activation by DLK is required for reactivation following interruption of the NGF-signaling pathway. Here, we find that forskolin and IL-1β-mediated reactivation also required DLK activity, further reinforcing the central role of DLK and JNK in reactivation of HSV from latency. DLK is known as a master regulator of a neuronal response to stress stimuli and mediates whole cell death, axon pruning, regeneration or degeneration depending on the nature of the stimuli. However, it has not before been linked to neuronal hyperexcitability or the response to IL-1β signaling. The known mechanisms of DLK activation include loss of AKT activation and phosphorylation by PKA (*Hao et al., 2016*; *Wu et al., 2015*), neither of which could be linked to HSV reactivation mediated by forskolin in this study. Following activation by DLK, one mechanism by which JNK is thought to permit lytic gene expression is via recruitment to viral promoters and histone phosphorylation. However, it is likely that there are additional, JNK-dependent effects including activation of pioneer or transcription factors that also mediate viral gene expression. Further insight into how HSV has hijacked this cellular pathway to induce lytic gene expression may lead to novel therapeutics that prevent reactivation, in addition to providing information on how viral gene expression initiates from promoters assembled into heterochromatin.

## Materials and methods

### Key resources table

| Reagent type (species) or resource | Designation | Source or reference | Identifiers | Additional information |
|---|---|---|---|---|
| Strain, strain background (*Mus musculus*, M/F) | CD1 | Charles River | Crl:CD1(ICR) | |
| Strain, strain background (*Human herpesvirus 1*) | HSV Us11-GFP | I gift from Ian Mohr, NYU. PMID:12915535 | | |
| Strain, strain background (*Human herpesvirus 1*) | HSV-1 17syn+ | A gift from Roger Everett, MRC Virology Unit Glasgow | | |
| Cell line (*Homo sapiens*) | 293LTV | Cell Biolabs | Cat # LTV-100 RRID:CVCL_JZ09 | |
| Cell line (*Cercopithecus aethiops*) | Vero | ATCC | Cat # CCCL-81 RRID:CVCL_0059 | |
| Recombinant DNA reagent | pCMV-VSV-G | A gift from Bob Weinberg/ Addgene PMID:12649500 | Cat # 8454 RRID:Addgene_8454 | |
| Recombinant DNA reagent | psPax2 | A gift from Didier Trono/Addgene | Cat # 12260 RRID:Addgene_12260 | |

*Continued on next page*

Continued

| Reagent type (species) or resource | Designation | Source or reference | Identifiers | Additional information |
|---|---|---|---|---|
| Antibody | Anti-phospho-Akt (S473) (Rabbit monoclonal) | Cell Signalling Technologies | Cat # 4060 RRID:AB_2315049 | WB (1:500) |
| Antibody | Anti-Akt (pan) (Rabbit monoclonal) | Cell Signalling Technologies | Cat # C67E7 RRID:AB_915783 | WB (1:1000) |
| Antibody | Anti-phopsho-c-Jun (Rabbit monoclonal) | Cell Signalling Technologies | Cat # 3270 RRID:AB_2129575 | WB (1:500) |
| Antibody | Anti-DLK/ MAP3K12 (Rabbit polyclonal) | Thermo Fisher | PA5-32173 RRID:AB_2549646 | WB (1:500) |
| Antibody | Anti-a-tubulin (Mouse monoclonal) | Millipore sigma | Cat # T9026 RRID:AB_477593 | WB (1:2500) |
| Antibody | Anti-Rabbit IgG Antibody (H+L), Peroxidase (Goat polyclonal) | Vector Labs | Cat # PI-1000 RRID:AB_2336198 | WB (1:10000) |
| Antibody | Anti-mouse IgG Antibody (H+L), Peroxidase (Horse polyclonal) | Vector Labs | Cat # PI-2000 RRID:AB_2336177 | WB (1:10000) |
| Antibody | Anti-H3K9me3S10P (Rabbit polyclonal) | Abcam | Cat # Ab5819 RRID:AB_305135 | IF (1:250) |
| Antibody | Anti-Beta-III Tubulin (Chicken polyclonal) | Millipore Sigma | Cat # AB9354 RRID:AB_570918 | IF (1:1000) |
| Antibody | Anti-γH2A.X (Mouse monoclonal) | Cell Signalling Technologies | Cat # 80312 RRID:AB_2799949 | IF (1:100) |
| Antibody | Anti-c-Fos (Rabbit polyclonal) | Novus | Cat # NB110-75039 RRID:AB_1048550 | IF (1:125) |
| Antibody | F(ab')2 Anti-Mouse IgG (H+L) Alexa Fluor 647, (Goat polyclonal) | Thermo Fisher | Cat # A21237 RRID:AB_2535806 | IF (1:1000) |
| Antibody | F(ab')2 Anti-Rabbit IgG (H+L) Alexa Fluor 555 (Goat polyclonal) | Thermo Fisher | Cat # A21425 RRID:AB_2535846 | IF (1:1000) |
| Antibody | Anti-Chicken IgY (H+L) Alexa Fluor 647 (Goat pAb) | Abcam | Cat # Ab150175 RRID:AB_2732800 | IF (1:1000) |
| Antibody | Anti-Chicken IgY (H+L) Alexa Fluor 488 (Goat polyclonal) | Abcam | Cat # Ab150173 RRID:AB_2827653 | IF (1:1000) |
| Antibody | F(ab')2 Anti-Rabbit IgG (H+L) Alexa Fluor 488 (Goat polyclonal) | Thermo Fisher | Cat # A-11070 RRID:AB_2534114 | IF (1:1000) |
| Antibody | Anti-Mouse IL-1R (Goat polyclonal) | Leinco Technologies | Cat # I-736 RRID:AB_2830857 | Blocking (2 ug/mL) |
| Sequence-based reagent | mGAP F | PMID:19515781 | PCR primers | CATGGCCTTCCGTGTGTTCCTA |

*Continued*

| Reagent type (species) or resource | Designation | Source or reference | Identifiers | Additional information |
|---|---|---|---|---|
| Sequence-based reagent | mGAP R | PMID:19515781 | PCR primers | GCGGCACGTCAGATCCA |
| Sequence-based reagent | ICP27 F | PMID:21285374 | PCR primers | GCATCCTTCGTGTTTGTCATTCTG |
| Sequence-based reagent | ICP27 R | PMID:21285374 | PCR primers | GCATCTTCTCTCCGACCCCG |
| Sequence-based reagent | ICP8 F | PMID:23322639 | PCR primers | GGAGGTGCACCGCATACC |
| Sequence-based reagent | ICP8 R | PMID:23322639 | PCR primers | GGCTTAAATCCGGCATGAC |
| Sequence-based reagent | ICP4 F | This paper | PCR primers | TGCTGCTGCTGTCCACGC |
| Sequence-based reagent | ICP4 R | This paper | PCR primers | CGGTGTTGACCACGATGAGCC |
| Sequence-based reagent | UL30 F | PMID:22383875 | PCR primers | CGCGCTTGGCGGGTATTAACAT |
| Sequence-based reagent | UL30 R | PMID:22383875 | PCR primers | TGGGTGTCCGGCAGAATAAAGC |
| Sequence-based reagent | UL48 F | This paper | PCR primers | TGCTCGCGAATGTGGTTTAG |
| Sequence-based reagent | UL48 R | This paper | PCR primers | CTGTTCCAGCCCTTGATGTT |
| Sequence-based reagent | gC F | This paper | PCR primers | CAGTTTGTCTGGTTCGAGGAC |
| Sequence-based reagent | gC R | This paper | PCR primers | ACGGTAGAGACTGTGGTGAA |
| Sequence-based reagent | shRNA: DLK-1 | Broad Institute: Genetic Perturbation Platform/Millipore Sigma | TRCN0000022573 | |
| Sequence-based reagent | shRNA: DLK-2 | Broad Institute: Genetic Perturbation Platform/Millipore Sigma | TRCN0000022572 | |
| Sequence-based reagent | shRNA: non-targeting control | PMID:16873256 | | |
| Commercial assay or kit | *Quick*-RNA Miniprep | Zymo Research | R1054 | |
| Commercial assay or kit | SuperScript IV First-Strand Synthesis System | ThermoFisher | 18091050 | |
| Commercial assay or kit | SYBR Green PCR Master Mix | ThermoFisher | 4309155 | |
| Chemical compound, drug | Acycloguanosine | Millipore Sigma | A4669 | 10 μM, 50 μM |
| Chemical compound, drug | FUDR | Millipore Sigma | F-0503 | 20 μM |
| Chemical compound, drug | Uridine | Millipore Sigma | U-3003 | 20 μM |
| Chemical compound, drug | SP600125 | Millipore Sigma | S5567 | 20 μM |

*Continued on next page*

*Continued*

| Reagent type (species) or resource | Designation | Source or reference | Identifiers | Additional information |
|---|---|---|---|---|
| Chemical compound, drug | GNE-3511 | Millipore Sigma | 533168 | 4 µM |
| Chemical compound, drug | GSK-J4 | Millipore Sigma | SML0701 | 2 µM |
| Chemical compound, drug | L-Glutamic Acid | Millipore Sigma | G5638 | 3.7 µg/mL |
| Chemical compound, drug | Forskolin | Tocris | 1099 | 60 µM |
| Chemical compound, drug | LY 294002 | Tocris | 1130 | 20 µM |
| Chemical compound, drug | 666–15 | Tocris | 5661 | 2 µM |
| Chemical compound, drug | SQ 22,536 | Tocris | 1435 | 50 µM |
| Chemical compound, drug | KT 5720 | Tocris | 1288 | 3 µM |
| Chemical compound, drug | TEA | Tocris | 3068 | 10 mM |
| Chemical compound, drug | CsCl | Tocris | 4739 | 3 mM |
| Chemical compound, drug | OG-L002 | Tocris | 6244 | 30 µM |
| Chemical compound, drug | S2101 | Tocris | 5714 | 20 µM |
| Chemical compound, drug | Tetrodotoxin | Tocris | 1069 | 1 µM |
| Chemical compound, drug | ESI-09 | Tocris | 4773 | 10 µM |
| Chemical compound, drug | ZD 7288 | Cayman | 15228 | 20 µM |
| Chemical compound, drug | 8-bromo-cyclic AMP | Cayman | 14431 | 125 µM |
| Chemical compound, drug | NGF 2.5S | Alomone Labs | N-100 | 50 ng/mL |
| Chemical compound, drug | Primocin | Invivogen | ant-pm-1 | 100 µg/mL |
| Chemical compound, drug | Aphidicolin | AG Scientific | A-1026 | 3.3 µg/mL |
| Chemical compound, drug | IL-1β | Shenendoah Bio. | 100–167 | 30 ng/mL |
| Chemical compound, drug | WAY-150138 | Pfizer, gift from Lynn Enquist and Jay Brown. | NA | 10 µg/mL |
| Chemical compound, drug | Fura-2 AM | Thermo Fisher | F1221 | 5 µM |
| Other | Hoescht Stain | Thermo | 62249 | 2 µM |

## Reagents

Compounds used in the study are as follows: Acycloguanosine, FUDR, Uridine, SP600125, GNE-3511, GSK-J4, L-glutamic acid, and Ivabradine (Millipore Sigma); Forskolin, LY 294002, 666–15, SQ 22536, KT 5720, tetraethylammonium chloride, cesium chloride, OG-L002, S2101, tetrotdotoxin, and ESI-09 (Tocris); 1,9-dideoxy-Forskolin, ZD 7288 and 8-bromo-cyclic AMP (Cayman Chemicals);

nerve growth factor 2.5S (Alomone Labs); Primocin (Invivogen); aphidicolin (AG Scientific); IL-1β (Shenandoah Biotechnology); WAY-150138 was kindly provided by Pfizer, Dr. Jay Brown at the University of Virginia, and Dr. Lynn Enquist at Princeton University. Compound concentrations were used based on previously published IC50s and assessed for neuronal toxicity using the cell body and axon health and degeneration index (*Supplementary file 1* Table 1 and 2). All compounds used had an average score ≤1. Untreated controls are quantified as 'Mock' treatments for all experiments.

## Preparation of HSV-1 virus stocks

HSV-1 stocks of eGFP-Us11 Patton were grown and titrated on Vero cells obtained from the American Type Culture Collection (Manassas, VA). Cells were maintained in Dulbecco's Modified Eagle's Medium (Gibco) supplemented with 10% FetalPlex (Gemini Bio-Products) and 2 mM L-Glutamine. Cells were confirmed to be mycoplasma negative using the Mycoplasma PCR Detection Kit (amb). eGFP-Us11 Patton (HSV-1 Patton strain with eGFP reporter protein fused to true late protein Us11 [*Benboudjema et al., 2003*]) was kindly provided by Dr. Ian Mohr at New York University.

## Primary neuronal cultures

Sympathetic neurons from the superior cervical ganglia (SCG) of post-natal day 0–2 (P0-P2) or adult (P21-P24) CD1 Mice (Charles River Laboratories) were dissected as previously described (*Cliffe et al., 2015*). Rodent handling and husbandry were carried out under animal protocols approved by the Animal Care and Use Committee of the University of Virginia (UVA). Ganglia were briefly kept in Leibovitz's L-15 media with 2.05 mM L-Glutamine before dissociation in Collagenase Type IV (1 mg/mL) followed by Trypsin (2.5 mg/mL) for 20 min each at 37°C. Dissociated ganglia were triturated, and approximately 10,000 neurons per well were plated onto rat tail collagen in a 24-well plate. Sympathetic neurons were maintained in CM1 (Neurobasal Medium supplemented with PRIME-XV IS21 Neuronal Supplement (Irvine Scientific), 50 ng/mL Mouse NGF 2.5S, 2 mM L-Glutamine, and Primocin). Aphidicolin (3.3 μg/mL), Fluorodeoxyuridine (20 μM) and Uridine (20 μM) were added to the CM1 for the first five days post-dissection to select against proliferating cells.

## Establishment and reactivation of latent HSV-1 infection in primary neurons

Latent HSV-1 infection was established in P6-8 and P30-32 sympathetic neurons from SCGs. Neurons were cultured for at least 24 hr without antimitotic agents prior to infection. The cultures were infected with eGFP-Us11 (Patton recombinant strain of HSV-1 expressing an eGFP reporter fused to true late protein Us11). Neurons were infected at a Multiplicity of Infection (MOI) of 7.5 PFU/cell (assuming $1.0 \times 10^4$ neurons/well/24-well plate) in DPBS +CaCl$_2$ +MgCl$_2$ supplemented with 1% Fetal Bovine Serum, 4.5 g/L glucose, and 10 μM Acyclovir (ACV) for 3 hr at 37 °C. Post-infection, inoculum was replaced with CM1 containing 50 μM ACV for 5–6 days, followed by CM1 without ACV. Reactivation was carried out in DMEM/F12 (Gibco) supplemented with 10% Fetal Bovine Serum, Mouse NGF 2.5S (50 ng/mL) and Primocin. Inhibitors were added either 1 hr prior to or concurrently with the reactivation stimulus. WAY-150138 (2–10 μg/mL) was added to reactivation cocktail to limit cell-to-cell spread. Reactivation was quantified by counting number of GFP-positive neurons or performing Reverse Transcription Quantitative PCR (RT-qPCR) of HSV-1 lytic mRNAs isolated from the cells in culture.

## Analysis of mRNA expression by reverse-transcription quantitative PCR (RT-qPCR)

To assess relative expression of HSV-1 lytic mRNA, total RNA was extracted from approximately $1.0 \times 10^4$ neurons using the Quick-RNA Miniprep Kit (Zymo Research) with an on-column DNase I digestion. mRNA was converted to cDNA using the SuperScript IV First-Strand Synthesis system (Invitrogen) using random hexamers for first-strand synthesis and equal amounts of RNA (20–30 ng/reaction). To assess viral DNA load, total DNA was extracted from approximately $1.0 \times 10^4$ neurons using the Quick-DNA Miniprep Plus Kit (Zymo Research). qPCR was carried out using *Power* SYBR Green PCR Master Mix (Applied Biosystems). The relative mRNA or DNA copy number was determined using the Comparative C$_T$ (ΔΔC$_T$) method normalized to mRNA or DNA levels in latently

infected samples. Viral RNAs were normalized to mouse reference gene GAPDH. All samples were run in duplicate on an Applied Biosystems QuantStudio 6 Flex Real-Time PCR System and the mean fold change compared to the reference gene calculated. Primers used are described in Key Resources Table.

## Western blot analysis
Neurons were lysed in RIPA Buffer with cOmplete, Mini, EDTA-Free Protease Inhibitor Cocktail (Roche) and PhosSTOP Phosphatase Inhibitor Cocktail (Roche) on ice for 1 hr with regular vortexing to aid lysis. Insoluble proteins were removed via centrifugation, and lysate protein concentration was determined using the Pierce Bicinchoninic Acid Protein Assay Kit (Invitrogen) using a standard curve created with BSA standards of known concentration. Equal quantities of protein (20–50 µg) were resolved on 4–20% gradient SDS-Polyacrylamide gels (Bio-Rad) and then transferred onto Polyvinylidene difluoride membranes (Millipore Sigma). Membranes were blocked in PVDF Blocking Reagent for Can Get Signal (Toyobo) for 1 hr. Primary antibodies were diluted in Can Get Signal Immunoreaction Enhancer Solution 1 (Toyobo) and membranes were incubated overnight at 4°C. HRP-labeled secondary antibodies were diluted in Can Get Signal Immunoreaction Enhancer Solution 2 (Toyobo) and membranes were incubated for 1 hr at room temperature. Blots were developed using Western Lightning Plus-ECL Enhanced Chemiluminescence Substrate (PerkinElmer) and ProSignal ECL Blotting Film (Prometheus Protein Biology Products) according to manufacturer's instructions. Blots were stripped for reblotting using NewBlot PVDF Stripping Buffer (Licor). Band density was quantified in ImageJ.

## Preparation of lentiviral vectors
Lentiviruses expressing shRNA against DLK (DLK-1 TRCN0000022572, DLK-2 TRCN0000022573), or a control lentivirus shRNA (*Everett et al., 2006*) were prepared by co-transfection with psPAX2 and pCMV-VSV-G (*Stewart et al., 2003*) using the 293LTV packaging cell line (Cell Biolabs). Supernatant was harvested at 40 hr and 64 hr post-transfection. Sympathetic neurons were transduced overnight in neuronal media containing 8 µg/mL protamine and 50 µM ACV.

## Immunofluorescence
Neurons were fixed for 15 min in 4% Formaldehyde and blocked in 5% Bovine Serum Albumin and 0.3% Triton X-100 and incubated overnight in primary antibody. Following primary antibody treatment, neurons were incubated for 1 hr in Alexa Fluor 488-, 555-, and 647-conjugated secondary antibodies for multi-color imaging (Invitrogen). Nuclei were stained with Hoechst 33258 (Life Technologies). Images were acquired using an sCMOS charge-coupled device camera (pco.edge) mounted on a Nikon Eclipse Ti Inverted Epifluorescent microscope using NIS-Elements software (Nikon). Images were analyzed and intensity quantified using ImageJ.

## Click-chemistry
For EdC-labeled HSV-1 virus infections, an MOI of 7.5 was used. EdC labeled virus was prepared using a previously described method (*McFarlane et al., 2019*). Click-chemistry was carried out a described previously (*Alandijany et al., 2018*) with some modifications. Neurons were washed with CSK buffer (10 mM HEPES, 100 mM NaCl, 300 mM Sucrose, 3 mM $MgCl_2$, 5 mM EGTA) and simultaneously fixed and permeabilized for 10 min in 1.8% methanol-free formaldehyde (0.5% Triton X-100, 1% phenylmethylsulfonyl fluoride (PMSF)) in CSK buffer, then washed twice with PBS before continuing to the click-chemistry reaction and immunostaining. Samples were blocked with 3% BSA for 30 min, followed by click-chemistry using EdC-labeled HSV-1 DNA and the Click-iT EdU Alexa Flour 555 Imaging Kit (ThermoFisher Scientific, C10638) according to the manufacturer's instructions. For immunostaining, samples were incubated overnight with primary antibodies in 3% BSA. Following primary antibody treatment, neurons were incubated for 1 hr in Alexa Fluor 488-, 555-, and 647-conjugated secondary antibodies for multi-color imaging (Invitrogen). Nuclei were stained with Hoechst 33258 (Life Technologies). Images were acquired at 60x using an sCMOS charge-coupled device camera (pco.edge) mounted on a Nikon Eclipse Ti Inverted Epifluorescent microscope using NIS-Elements software (Nikon). Images were analyzed and intensity quantified using ImageJ.

## Cytosolic Ca²⁺ imaging using ratiometric Fura-2 (microscopy)

For ratiometric $Ca^{2+}$ imaging, neurons were seeded on coverslips and incubated for 30 min at RT with 5 µM Fura-2-AM, 0.02% pluronic acid in Ringer solution (in mM, 155 NaCl, 4.5 KCl, 2 $CaCl_2$, 1 $MgCl_2$, 5 HEPES, 10 glucose, adjusted to pH 7.4). Excitations of Fura-2 at 340 nm and 380 nm emissions were carried out using a DG4 Illuminator (Sutter Instruments). Emissions were collected at 510 nm using an ORCA-Flash 4.0 V2 CMOS camera (Hamamatsu). Cells were imaged every 500 milliseconds for the duration of the experiment. Acetylcholine (100 µM) and Ionomycin (2 µM) were applied at indicated timepoints. Data were acquired and processed using SlideBook six software.

## Cytosolic Ca²⁺ imaging using ratiometric Fura-2 (FlexStation)

For ratiometric $Ca^{2+}$ imaging, neurons were seeded on a 96-well black walled plate and incubated for 30 min at RT with 5 µM Fura-2-AM, 0.02% of pluronic acid in Ringer solution ([in mM] 155 NaCl, 4.5 KCl, 10 $CaCl_2$, 1 $MgCl_2$, 5 HEPES, 10 glucose, pH 7.4). Fura-2 emissions were collected at 510 nm and with 340/380 nm excitation. Plates were imaged using the FlexStation 3 (Molecular Devices). Cells were imaged every 5 s for the duration of the experiment.

### Statistical analysis

Power analysis was used to determine the appropriate sample sizes for statistical analysis. All statistical analysis was performed using Prism V8.4. An unpaired t-test was used for all experiments where the group size was 2. All other experiments were analyzed using a one-way ANOVA with a Tukey's multiple comparison. Specific analyses are included in the figure legends. For all reactivation experiments measuring GFP expression, viral DNA, gene expression or DNA load, individual biological replicates were plotted (an individual well of primary neurons) and all experiments were repeated from pools of neurons from at least three litters. EdC virus and H3K9me3S10/p co-localization was quantified using ImageJ after sample blinding of at least 8 fields of view from two biological replicates. Mean fluorescence intensity of γH2AX and H3K9me3pS10 was quantified using ImageJ from at least 100 cells from at least three biological replicates.

## Acknowledgements

We thank Ian Mohr (NYU) for supplying the Us11-GFP virus used in this study, Roger Everett for the HSV-1 17syn+ and Jay Brown (UVA) and Lynn Enquist (Princeton University) for the generous gift of WAY-150138. This work was supported by NIH/NINDS R01NS105630 (to ARC), NIH/NIAID T32AI007046 (SRC and JBS), NIH/NEI F30EY030397 (JBS), NIH/NIGMS T32GM008136 (SD), T32GM007267 (JBS), NIH GM108989 (BND), NIH T32 GM007055 (PVS), and MRC (https://mrc.ukri.org) MC_UU_12014/5 (CB). psPAX2 was a gift from Didier Trono (Addgene plasmid # 12260; http://n2t.net/addgene:12260; RRID:Addgene_12260) and pCMV-VSV-G was a gift from Bob Weinberg (Addgene plasmid # 8454; http://n2t.net/addgene:8454; RRID:Addgene_8454). We also thank the reviewers for their comments on the manuscript.

## Additional information

### Funding

| Funder | Grant reference number | Author |
| --- | --- | --- |
| National Institute of Neurological Disorders and Stroke | R01NS105630 | Anna R Cliffe |
| National Institute of Allergy and Infectious Diseases | T32AI007046 | Sean R Cuddy<br>Jon Suzich |
| National Institute of General Medical Sciences | T32GM008136 | Sara Dochnal |
| National Institute of General Medical Sciences | T32GM007267 | Jon Suzich |
| National Eye Institute | F30EY030397 | Jon Suzich |
| Medical Research Council | MC_UU_12014/5 | Chris Boutell |

| National Institute of General Medical Sciences | GM108989 | Bimal N Desai |
| --- | --- | --- |
| National Institute of General Medical Sciences | GM007055 | Philip V Seegren |

The funders had no role in study design, data collection and interpretation, or the decision to submit the work for publication.

### Author contributions

Sean R Cuddy, Data curation, Investigation, Visualization, Methodology, Writing - review and editing; Austin R Schinlever, Sara Dochnal, Jon Suzich, Investigation, Methodology, Writing - review and editing; Philip V Seegren, Data curation, Investigation, Writing - original draft, Writing - review and editing; Parijat Kundu, Taylor K Downs, Investigation; Mina Farah, Formal analysis; Bimal N Desai, Resources, Supervision, Validation; Chris Boutell, Resources, Methodology, Writing - review and editing; Anna R Cliffe, Conceptualization, Resources, Data curation, Formal analysis, Supervision, Funding acquisition, Validation, Investigation, Visualization, Methodology, Writing - original draft, Project administration, Writing - review and editing

### Author ORCIDs

Sean R Cuddy https://orcid.org/0000-0002-2062-0691
Austin R Schinlever http://orcid.org/0000-0003-3401-0904
Jon Suzich http://orcid.org/0000-0002-6087-2893
Parijat Kundu http://orcid.org/0000-0003-1944-4579
Bimal N Desai http://orcid.org/0000-0002-3928-5854
Anna R Cliffe https://orcid.org/0000-0003-1136-5171

### Ethics

Animal experimentation: This study was performed in strict accordance with the recommendations in the Guide for the Care and Use of Laboratory Animals of the National Institutes of Health. Rodent handling and husbandry were carried out under animal protocols approved by the Animal Care and Use Committee of the University of Virginia (UVA). All of the animals were handled according to approved institutional animal care and use committee (IACUC) protocols (#4134) of the University of Virginia.

### Decision letter and Author response

Decision letter https://doi.org/10.7554/eLife.58037.sa1
Author response https://doi.org/10.7554/eLife.58037.sa2

## Additional files

### Supplementary files

• Supplementary file 1. Table 1. Cell Body Score for Neuronal Health and Degeneration IndexScoring system used to determine neuronal health based on morphology of the soma following treatment with compounds used in this study. Table 2. Axon Score for Neuronal Health and Degeneration IndexScoring system used to determine neuronal health based on morphology of the axons following treatment with compounds used in this study.

• Transparent reporting form

### Data availability

All data generated or analysed during this study are included in the manuscript and supporting files. Source data files have been provided for all figures.

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
