## [Decision Letter]

**Acceptance summary:**

Herpes simplex virus type 1 (HSV1) establishes lifelong latent infections in neurons in the peripheral nervous system and periodically reactivates to produce new infectious virions and infect new hosts. Triggers of HSV1 reactivation are well recognized, however, the underlying molecular mechanisms have not been fully elucidated. This manuscript could demonstrate that the interleukin IL-1β could trigger HSV-1 reactivation and thereby provides novel insights into the fundamental mechanisms of HSV reactivation.

**Decision letter after peer review:**

Thank you for submitting your article "Neuronal hyperexcitability is a DLK-dependent trigger of HSV-1 reactivation that can be induced by IL-1" for consideration by *eLife*. Your article has been reviewed by three peer reviewers, and the evaluation has been overseen by a Reviewing Editor and Päivi Ojala as the Senior Editor. The reviewers have opted to remain anonymous.

The reviewers have discussed the reviews with one another and the Reviewing Editor has drafted this decision to help you prepare a revised submission.

Summary:

This work addresses the key question how the herpesvirus HSV-1 reactivates from latency in neurons and shows that neuronal excitability plays a major role for controlling latency and reactivation. How hyperexcitability might influence the behavior of a latent, neurotrophic virus was previously unknown, and the authors show that neuronal hyperexcitability induces HSV reactivation in a DLK/*JNK*-dependent manner. In additon, the authors identify the cytokine IL-1β as a stimulus that triggers HSV reactivation in neurons, dependent on neuronal excitability, which is also a novel finding and of great interest for the field.

Essential revisions:

The reviewers all agree that your work about the potential link between IL1β, neuronal hyperexcitability, and HSV-1 reactivation is very interesting. However, we think that the three experiments listed below would be needed to substantiate the conclusion regarding the link between these three elements.

1) To make sure that the results obtained with the inhibitors are not off-target effects, experiments with KO cells or siRNA knockdowns of DLK/*JNK* would strengthen the manuscript. Can you please specify in the manuscript the specific targets of the three inhibitors that were used – if they have discrete mechanisms of action, off-target effects may actually not be a problem. However, KO or knockdown experiments would validate the inhibitor results by an independent method and should be doable in these cultures.

2) To substantiate the very interesting finding with IL1β, an experiment with a neutralizing IL1β antibody should be performed to unequivocally show that IL1β induces reactivation (this would exclude that impurities in the cytokine batch such as LPS activate the cells).

3) To unequivocally show that IL1β induces reactivation through increasing neuronal hyperexcitability, calcium flux, which is induced by neuronal hyperexcitability, should be measured. A simple method to do this would be the use of Fura-2 AM or similar dyes. An advantage of this approach is that it could be measured what percentage of neurons are excited upon IL1β treatment and this could be correlated with the percentage of neurons that reactivate. This could also be performed in the presence of IL1β neutralizing antibodies to confirm that this cytokine induces neuronal hyperexcitability and HSV-1 reactivation.

These three additional experiments would make the report more robust and elegantly correlate hyperexcitability of neurons with HSV-1 reactivation.

Reviewer #1:

The manuscript by Cuddy et al., entitled "Neuronal hyperexcitability is a DLK-dependent trigger of HSV-1 reactivation that can be induced by IL-1" aims at discovering and characterizing novel stimuli that lead to reactivation of herpes simplex virus (HSV).

They show that reactivation can occur upon addition of compounds that induce neuronal excitability and that inhibitors of this process reduce reactivation. They also show that interleukin 1 β (IL1β) induces HSV reactivation, linked it to neuronal excitability and suggest that this may be a reactivation trigger in vivo during situations in which IL1β expression is enhanced (e.g., fever, stress).

The work is very interesting, solid and well performed but it requires certain controls and experiments to support the claims of the authors.

1) The authors indicate that the compounds used induce neuronal hyperexcitability and that this is demonstrated by the detection of DLK, gH2AX, indicative of DNA damage, and H3K9me/S10p. These are indirect readouts of neuronal excitability. To prove that there is such excitability the authors should perform other assays, such as patch clamp. This is of particular importance to prove that IL1β-induced reactivation is mediated by induction of neuronal excitability.

2) Similarly, the authors base most of their conclusions on the use of inhibitors of certain pathways. This is a valid approach but it will be better to complement it with KD or KO experiments due to the potential unspecific effect of the inhibitors, especially if the concentration used is much higher than their Ki (e.g., 4 μm of GNE-3511 when its Ki is 0.5 nM). For instance ZD 7288 is not a specific inhibitor of HCN channels since it also blocks Na currents (Wu et al., 2012). They should use KD or KO at least for experiments dealing with the effect of IL1β in reactivation and its potential link to DLK. There are published reports on other topics using KD and KO for IL1β (or its receptor) and DLK KD and conditional KO.

3) Regarding IL1β, there are not data in the manuscript showing how active the recombinant protein used is. It would be important to test the activity of IL1β in a classic assay such as proliferation to know the amount of units that are needed to induce reactivation and whether this would correspond to physiological levels. Moreover, since some of these recombinant proteins may contain impurities that activate other signaling cascades, it is important to know the specificity of IL1β signaling in this process. The authors could use neutralizing antibodies blocking IL1β or its receptor (or KD/KO as above) and then measure HSV reactivation. What is the level of IL1β receptor in the neurons used in these experiments?

4) The use of imaging techniques to show histone modifications should be complemented by ChIP assays, especially when investigating the effect of IL1β and neuronal hyperexcitability and reactivation. They should investigate these modifications at least on control cellular genes and HSV IE and E genes upon addition of IL1β and conditions of neuronal excitability. The authors are experts on this type of assay.

Reviewer #2:

This manuscript focuses on the hypothesis that reactivation of HSV in cultures of primary neonatal mouse sympathetic neurons is triggered by neuronal hyperexcitibility. Understanding the triggers and signaling pathways that activate the latent HSV genome in the individual latently infected neuron remains an important and challenging question. The authors utilize a mouse superior cervical ganglion neuronal culture model of short term latency. Latency is forced through suppression of viral replication using the antiviral DNA chain terminator acyclovir (ACV) and the cultures are infected with an moi of 7.5 pfu/cell in its presence. After 6 days, ACV is removed and two days later, these cultures are considered latently infected and reactivation triggers and inhibitors are applied. In this model, activation occurs in ~2% of the neurons, presumably 100% latently infected. Thus even in this culture system, a minor subset of the neurons actually undergo reactivation despite the application of the compounds to all of the neurons in the well, making analysis at the individual neuron level critical but absent in this study in certain key analyses.

1) The stated aim of this study is to "…. identify novel triggers of HSV reactivation and determine if they involved a bi-phasic mode of reactivation." The authors utilize a series of antagonists, agonists, and inhibitors on the cultures. These authors confirm previous studies by others using this model showing that forskolin, like NGF withdrawal, results in the activation of the viral lytic cycle. There are several novel observations that are of interest. However, it is this reviewer's opinion that in the absence of any measurement of excitability in these neurons (cultures) is problematic considering the impact that ongoing HSV infection may have on the physiological properties of the neurons and their responses to commonly used agents. There is the assumption that the electrophysiological effect of the agents added to the cultures is totally predictable. This is not so straightforward. For example, it has been reported that forskolin reduces excitability of SCG neurons by enhancing the spike frequency-dependent adaptation. FSK reduced the number of spikes from 8 to 2. doi.org/10.1371/journal.pone.0126365. While an argument for a role of hyperexcitibility can be made from the studies, the strong stance taken throughout the manuscript and including the title is not warranted. While the authors' observations are consistent with the connection between neuronal excitation and reactivation, this has been shown in 2003 in a similar culture model in which the calcium-dependent activation of VR-1 channel resulted in HSV reactivation. "A common feature of all of these reactivation stimuli (including forskolin) is hyperexcitation of the neuron resulting in elevated calcium cause by VR-1 activation." doi 10.1099/vir.0.18828-0.

2) In this study, advances in drawing links between the viral genome and histone modifications that could potentially be associated with the observed increase in transcriptional activity from the genome (phospohmethyl switch) have been made but fall short of showing a connection to reactivation. While "reactivation" is loosely used throughout the manuscript, it is a term that has a meaning and is based on infectious virus production from a previously latent viral genome. The use of Us11GFP as a surrogate marker is understandable and acceptable. However, reactivation is not the detection of transcriptional fluctuations from the viral genome. This is a well-recognized feature of in vivo viral latency. This low level of virally related RNA is not necessarily limited to gene coding regions and not associated with detectable viral protein expression or infectious virus (ie not associated with reactivation). In this culture model whether changes in transcripts observed in phase I are required for Phase II is not clear. The localization of these Phase I transcripts is not known nor do we know their magnitude. Do the phase I transcripts localize to neurons in which the viral genome is associated with HeK9me3/S10p? Are these the neurons that proceed to reactivation? The potential biological significance of the data could be better appreciated if the authors tied these changes together. It is difficult to appreciate the biological relevance of the viral RNA detected since it is presented as a value normalized to the levels in control cultures prior to induction. The level of virally related RNA present in the cultures prior to induction is important information. This data presentation style is also an issue with the number of EdC labelled neurons in the culture. Assuming that the cultures were infected at the same moi as US11GFP virus, all of the neurons would be expected to contain labelled HSV DNA. The % of the neurons in the cultures that contain EdC labelled viral DNA should be included but only normalized values are given.

3) The wt superinfection experiment is quite interesting and seems to reveal that only a small portion of the neurons in the culture either contain the viral genome or contain reactivation competent genomes. Presumably, there are ~10,000 neurons/well. The infected cultures were superinfected with wt virus at an moi of 10. This would result in all neurons being superinfected, the vast majority with multiple wild type virions. Presumably this is known. The expectation would be that the wt virus would activate the US11GFP genome and most or all of the neurons would express GFP. Variations on this type of superinfection experiment have been done many times and reported in the literature. It is surprising that only ~650 of the 10,000 neurons expressed GFP. Does this suggest that most of the neurons do not contain reactivation competent "latent" DNA and what are the implications of this?

Reviewer #3:

Using a powerful neuronal culture model of HSV latency and reactivation, Cuddy et al. establish how fundamental parameters of neuronal cell physiology regulate critical aspects of the virus latency program at the molecular, mechanistic level. The work addresses a key, long-standing knowledge gap that has remained intractable for some time by directly addressing the role of neuronal excitability and its role in controlling latency and reactivation. While neuron hyperexcitability enables firing of repeated action potentials, how this basic property of an excitable host cell type might influence the behavior of a resident latent, neurotrophic virus was unknown. The authors definitively establish that neuronal hyperexcitability, modeled by forskolin or cAMP mimetic treatment, induces HSV reactivation in a DLK/*JNK*-dependent manner. This indicates that latent HSV1 responds to both axonal stress signaling and prolonged hyperexcitability via a common pathway (DLK/*JNK*) to result in reactivation. They go on to show that neuronal hyperexcitability and neuronal activity trigger reactivation via a histone phospho/methyl switch (H3K9me3/S10p), which enables Phase I viral gene expression from epigenetically silenced genomes containing repressive histone methylation marks. Most significantly, the authors' establish that reactivation dependent upon DLK/*JNK* was induced by (i) forskolin and obviated by blocking hyperpolarization-activated cyclic nucleotide gated (HCN) channels that are normally activated by membrane hyper-polarization; (ii) chronic neuronal excitability in response to extracellular [KCl]; and (iii) direct induction of neuronal hyperexcitability by release from a tetrodotoxin (TTX) block. Finally, they identify IL-1β, a known mediator of prolonged stress inflammation, and neuronal hyperexcitability, as a physiological stimulus that triggers HSV reactivation in mature neurons. IL1β induced reactivation was suppressed by TTX, an inhibitor of voltage gated sodium channels, indicating that IL1β triggers reactivation by increasing neuronal activity. Reactivation induced by IL1β was similarly dependent upon DLK/*JNK* signaling via a histone phospho/ methyl switch that triggers viral genome wide expression (Phase 1).

The experiments are well executed, the data are appropriately rigorous and convincing and the manuscript is well written. The study makes important, exciting findings by answering long-standing, highly significant questions regarding molecular mechanisms that have been nearly impossible to investigate in small animal models. It will undoubtedly have an oversized impact on the field and provides fundamental insight on understanding how a neurotrophic pathogen colonizes a host neuron, remains in an epigenetically silenced state, and periodically re-emerges to replicate productively in response to neuronal excitability. It will be of interest to a wide readership interested in neuronal cell biology, infection biology, stress responses and the epigenetic control of gene expression.

I have no further experiments to suggest that will improve this already terrific study.

---

## [Author Response]

Essential revisions:The reviewers all agree that your work about the potential link between IL1β, neuronal hyperexcitability, and HSV-1 reactivation is very interesting. However, we think that the three experiments listed below would be needed to substantiate the conclusion regarding the link between these three elements.1) To make sure that the results obtained with the inhibitors are not off-target effects, experiments with KO cells or siRNA knockdowns of DLK/JNK would strengthen the manuscript. Can you please specify in the manuscript the specific targets of the three inhibitors that were used – if they have discrete mechanisms of action, off-target effects may actually not be a problem. However, KO or knockdown experiments would validate the inhibitor results by an independent method and should be doable in these cultures.

We have added more details on the published specificity of the DLK inhibitor used. We have also validated the DLK inhibitor data by knock down with 2 independent shRNAs and shown that this inhibits HSV reactivation and Phase I gene expression following forskolin treatment, in addition to reactivation in response to release from a tetrodotoxin block and following IL-1 treatment. Knock-down of JNK itself is more challenging because there are three *JNK* encoding genes expressed in sympathetic neurons. However, because JNK is activated by DLK in neurons we are convinced that the combined data using JNK and DLK inhibitors in addition to the DLK knock-down experiments demonstrate a requirement for DLK in hyperexcitability-induced HSV reactivation.

We have also added more information on the mechanisms of action of the three HCN channel inhibitors. We have not performed knockdown of the HCN channels because there are four possible HCN channels expressed in sympathetic neurons and we have been unable to be sure of the specificity of available antibodies to know which are present. This is why we confirmed the data using the HCN channel inhibitor (ZD7288) with two additional HCN channel blockers. However, we have also changed the wording of the manuscript to acknowledge that ZD7288 can also inhibit voltage gated sodium channel activity.

2) To substantiate the very interesting finding with IL1β, an experiment with a neutralizing IL1β antibody should be performed to unequivocally show that IL1β induces reactivation (this would exclude that impurities in the cytokine batch such as LPS activate the cells).

We thank the reviewers for drawing our attention to the possibility of impurities in the IL-1β. To confirm that our observed phenotypes are linked to IL-1 specifically and not impurities in the cytokine batch, we have performed experiments with an IL1 RI neutralizing antibody. We have found that this inhibits reactivation in response to IL-1 treatment in addition to inhibiting histone phosphorylation and IL-1 mediated increased neuronal excitability.

3) To unequivocally show that IL1β induces reactivation through increasing neuronal hyperexcitability, calcium flux, which is induced by neuronal hyperexcitability, should be measured. A simple method to do this would be the use of Fura-2 AM or similar dyes. An advantage of this approach is that it could be measured what percentage of neurons are excited upon IL1β treatment and this could be correlated with the percentage of neurons that reactivate. This could also be performed in the presence of IL1β neutralizing antibodies to confirm that this cytokine induces neuronal hyperexcitability and HSV-1 reactivation.

We have now carried out calcium flux experiments in IL-1 treated neurons and demonstrated that sympathetic neurons exhibit increased intracellular calcium flux following stimulation when pre-treated with IL-1β. This response was inhibited by co-treatment with the IL-1R blocking antibody.

The reviewers do bring up an interesting point concerning the fraction of neurons capable of undergoing reactivation with IL-1 treatment. We also performed additional experiments showing that the fraction of neurons reactivating with IL-1 is comparable to the fraction reactivating with forskolin treatment. The ability of only a subpopulation of latently infected neurons to undergo reactivation could be due to heterogeneity in neuronal excitation, which we do observe based on the calcium imaging. However, there are likely many factors that contribute to reactivation efficiency including heterogeneity in viral latent gene expression and epigenetic structure of the latent viral genome.

These three additional experiments would make the report more robust and elegantly correlate hyperexcitability of neurons with HSV-1 reactivation.Reviewer #1:The manuscript by Cuddy et al., entitled "Neuronal hyperexcitability is a DLK-dependent trigger of HSV-1 reactivation that can be induced by IL-1" aims at discovering and characterizing novel stimuli that lead to reactivation of herpes simplex virus (HSV).They show that reactivation can occur upon addition of compounds that induce neuronal excitability and that inhibitors of this process reduce reactivation. They also show that interleukin 1 β (IL1β) induces HSV reactivation, linked it to neuronal excitability and suggest that this may be a reactivation trigger in vivo during situations in which IL1β expression is enhanced (e.g., fever, stress).The work is very interesting, solid and well performed but it requires certain controls and experiments to support the claims of the authors.1) The authors indicate that the compounds used induce neuronal hyperexcitability and that this is demonstrated by the detection of DLK, gH2AX, indicative of DNA damage, and H3K9me/S10p. These are indirect readouts of neuronal excitability. To prove that there is such excitability the authors should perform other assays, such as patch clamp. This is of particular importance to prove that IL1β-induced reactivation is mediated by induction of neuronal excitability.

We have now carried out calcium flux experiments in IL-1 treated neurons and demonstrated that sympathetic neurons exhibit increased intracellular calcium flux following stimulation when pre-treated with IL-1β.

2) Similarly, the authors base most of their conclusions on the use of inhibitors of certain pathways. This is a valid approach but it will be better to complement it with KD or KO experiments due to the potential unspecific effect of the inhibitors, especially if the concentration used is much higher than their Ki (e.g., 4 μm of GNE-3511 when its Ki is 0.5 nM). For instance ZD 7288 is not a specific inhibitor of HCN channels since it also blocks Na currents (Wu et al., 2012). They should use KD or KO at least for experiments dealing with the effect of IL1β in reactivation and its potential link to DLK. There are published reports on other topics using KD and KO for IL1β (or its receptor) and DLK KD and conditional KO.

To confirm the link to DLK we have performed DLK shRNA knock-down experiments using 2 independent shRNAs and found that this inhibits HSV reactivation and Phase I gene expression following forskolin treatment, in addition to reactivation in response to release from a tetrodotoxin block and following IL-1 treatment.

We have also amended the text in regards to ZD 7288.

3) Regarding IL1β, there are not data in the manuscript showing how active the recombinant protein used is. It would be important to test the activity of IL1β in a classic assay such as proliferation to know the amount of units that are needed to induce reactivation and whether this would correspond to physiological levels. Moreover, since some of these recombinant proteins may contain impurities that activate other signaling cascades, it is important to know the specificity of IL1β signaling in this process. The authors could use neutralizing antibodies blocking IL1β or its receptor (or KD/KO as above) and then measure HSV reactivation. What is the level of IL1β receptor in the neurons used in these experiments?

The IL-1β used I has ED50 = 1.09 pg/mL based on proliferation of D10S cells, according to the certificate of analysis. This is equivalent to 9.2 x 10^8^ units/mg. The units have been added into the Materials and methods. The reviewer does raise an interesting point concerning exposure of neurons to IL-1 and local versus systemic concentrations. To our knowledge, the serum concentration of IL-1β during fever is less than the concentration of recombinant IL-1β used here (1-1000pg/ml compared to 30ng/ml here). However, it is harder to assess the physiological level of IL-1 exposure to a neuron, especially since IL-1α/β can be release in high levels by epithelial cells, which will be in close proximity to neurites. In addition, satellite glial cells, which are found in close proximity to the somas of peripheral neurons are known producers of IL-1. Therefore, the exact physiological concentration of IL-1 to which a neuron can be exposed is incredibly challenging to assess.

We thank the reviewer for drawing our attention to the possibility of impurities in the IL-1β. To confirm that our observed phenotypes are linked to IL-1 specifically and not impurities in the cytokine batch, we have performed experiments with an anti-ILR blocking antibody. We have found that this inhibits reactivation in response to IL-1 treatment in addition to inhibiting histone phosphorylation and IL-1 mediated increased neuronal excitability.

4) The use of imaging techniques to show histone modifications should be complemented by ChIP assays, especially when investigating the effect of IL1β and neuronal hyperexcitability and reactivation. They should investigate these modifications at least on control cellular genes and HSV IE and E genes upon addition of IL1β and conditions of neuronal excitability. The authors are experts on this type of assay.

While we are experts at performing ChIP assays, each replicate requires around 30 mice and needs repeating about 5 times and would likely need multiple time points. The imaging approach gives valuable information on the co-localization of viral genomes with the histone phospho/methyl switch on a single neuron basis and accounts for the heterogeneity in the viral genome epigenetic structure. In the future we hope to perform additional studies on the epigenetic changes over the course of hyperexcitability-mediated reactivation and whether we can develop novel approaches that probe the nature of the heterogeneity. We hope that the reviewer appreciates the scale of these experiments.

Reviewer #2:This manuscript focuses on the hypothesis that reactivation of HSV in cultures of primary neonatal mouse sympathetic neurons is triggered by neuronal hyperexcitibility. Understanding the triggers and signaling pathways that activate the latent HSV genome in the individual latently infected neuron remains an important and challenging question. The authors utilize a mouse superior cervical ganglion neuronal culture model of short term latency. Latency is forced through suppression of viral replication using the antiviral DNA chain terminator acyclovir (ACV) and the cultures are infected with an moi of 7.5 pfu/cell in its presence. After 6 days, ACV is removed and two days later, these cultures are considered latently infected and reactivation triggers and inhibitors are applied. In this model, activation occurs in ~2% of the neurons, presumably 100% latently infected. Thus even in this culture system, a minor subset of the neurons actually undergo reactivation despite the application of the compounds to all of the neurons in the well, making analysis at the individual neuron level critical but absent in this study in certain key analyses.1) The stated aim of this study is to "…. identify novel triggers of HSV reactivation and determine if they involved a bi-phasic mode of reactivation." The authors utilize a series of antagonists, agonists, and inhibitors on the cultures. These authors confirm previous studies by others using this model showing that forskolin, like NGF withdrawal, results in the activation of the viral lytic cycle. There are several novel observations that are of interest. However, it is this reviewer's opinion that in the absence of any measurement of excitability in these neurons (cultures) is problematic considering the impact that ongoing HSV infection may have on the physiological properties of the neurons and their responses to commonly used agents. There is the assumption that the electrophysiological effect of the agents added to the cultures is totally predictable. This is not so straightforward. For example, it has been reported that forskolin reduces excitability of SCG neurons by enhancing the spike frequency-dependent adaptation. FSK reduced the number of spikes from 8 to 2. doi.org/10.1371/journal.pone.0126365. While an argument for a role of hyperexcitibility can be made from the studies, the strong stance taken throughout the manuscript and including the title is not warranted. While the authors' observations are consistent with the connection between neuronal excitation and reactivation, this has been shown in 2003 in a similar culture model in which the calcium-dependent activation of VR-1 channel resulted in HSV reactivation. "A common feature of all of these reactivation stimuli (including forskolin) is hyperexcitation of the neuron resulting in elevated calcium cause by VR-1 activation." doi 10.1099/vir.0.18828-0.

We have amended the text to reflect the reviewer’s comments with regarding forskolin. However, we do use multiple inhibitors that all function to block neuronal excitation. In addition to forskolin, we show that reactivation can be directly induced via release from a tetrodotoxin block and potassium chloride.

We have also added addition data demonstrating that IL-1 treatment induces an increase in neuronal firing.

2) In this study, advances in drawing links between the viral genome and histone modifications that could potentially be associated with the observed increase in transcriptional activity from the genome (phospohmethyl switch) have been made but fall short of showing a connection to reactivation. While "reactivation" is loosely used throughout the manuscript, it is a term that has a meaning and is based on infectious virus production from a previously latent viral genome. The use of Us11GFP as a surrogate marker is understandable and acceptable. However, reactivation is not the detection of transcriptional fluctuations from the viral genome. This is a well-recognized feature of in vivo viral latency.

The reviewer is correct that the phosho/methyl switch occurs prior to full reactivation of the virus in this system. In addition, the reviewer brings up an interesting point that fluctuations do occur in latency in vivo in mouse models. We predict that such fluctuations are linked to the state of the neuron; for example, Ma et al., 2014, found that latent neurons undergoing fluctuations in lytic gene expression had increased expression of *Bim*, a gene known to be induced following activation of DLK/JNK in neurons. Therefore, it is possible that small fluctuations in lytic gene expression or abortive reactivation does occur in vivo and could be associated with JNK activity and potentially a phospho/methyl switch.

Because of the repressed state of the viral genome in latency, we predict that there are likely many changes that occur on the viral genome prior to full packaging in the viral particle (and hence fully reactivating). We hope that the reviewer appreciates the value in understanding the early changes that occur on the viral genome following a reactivation stimulus.

This low level of virally related RNA is not necessarily limited to gene coding regions and not associated with detectable viral protein expression or infectious virus (ie not associated with reactivation). In this culture model whether changes in transcripts observed in phase I are required for Phase II is not clear. The localization of these Phase I transcripts is not known nor do we know their magnitude. Do the phase I transcripts localize to neurons in which the viral genome is associated with HeK9me3/S10p? Are these the neurons that proceed to reactivation? The potential biological significance of the data could be better appreciated if the authors tied these changes together.

The role of a general opening of viral chromatin and production of mRNA might not necessarily be related to protein synthesis. There are many unanswered questions on the progression to full viral reactivation that will require multiple future studies and we hope the reviewer and others can use our study to build on and answer these important and challenging questions.

It is difficult to appreciate the biological relevance of the viral RNA detected since it is presented as a value normalized to the levels in control cultures prior to induction. The level of virally related RNA present in the cultures prior to induction is important information.

The use of relative expression is standard from multiple studies in the field. The values for our mock treated, latently infected neurons are close to the limit of detection of our assay. In terms of gene expression, mRNA levels are relative at any one time and even silenced genes are unlikely to be fully silencing in all cells all the time.

This data presentation style is also an issue with the number of EdC labelled neurons in the culture. Assuming that the cultures were infected at the same moi as US11GFP virus, all of the neurons would be expected to contain labelled HSV DNA. The % of the neurons in the cultures that contain EdC labelled viral DNA should be included but only normalized values are given.

The normalized values are given so that the proportion of viral genomes that are co-localizing with the histone phospho-methyl switch. When working with primary neuronal systems there is a great deal of heterogeneity including the proportion of latently infected neurons from one field of view to another. This is why we perform a high number of replicates in all our experiments. However, this does require quantification of the proportion of latently infected neurons using click-chemistry where multiple images through the Z-plane of the neurons are required to find genomes the 3-dimensional neuronal nucleus. This is much more challenging than in fibroblasts. For this reason, we felt that the proportion of neurons that we imaged in an unbiased fashion that can be found co-localized with the phospho/methyl switch would be the best approach.

3) The wt superinfection experiment is quite interesting and seems to reveal that only a small portion of the neurons in the culture either contain the viral genome or contain reactivation competent genomes. Presumably, there are ~10,000 neurons/well. The infected cultures were superinfected with wt virus at an moi of 10. This would result in all neurons being superinfected, the vast majority with multiple wild type virions. Presumably this is known. The expectation would be that the wt virus would activate the US11GFP genome and most or all of the neurons would express GFP. Variations on this type of superinfection experiment have been done many times and reported in the literature. It is surprising that only ~650 of the 10,000 neurons expressed GFP. Does this suggest that most of the neurons do not contain reactivation competent "latent" DNA and what are the implications of this?

Yes, this does mean that many neurons do not contain reactivation competent genomes, with more reactivation occurring in response to superinfection compared to forskolin. This is not unique to our model and model systems using axotomy or in vivo reactivation also exhibit low levels of reactivation (and for heat-shock in vivo between 50-65 neurons in total have been reported as reactivating; Thompson et al., 2009). It is not surprising that superinfection results in increased reactivation compared to forskolin. However, the numbers for superinfection do indicate that there could be genomes that are unable to undergo reactivation. As mentioned above, quantifying the exact number of infected neurons is challenging, but based on the fields of view we have imaged viral genomes appear to be present in 40-50% of neurons. The number of surviving neurons after dissection is approximately 5,000-7,500. Therefore, the number of latently infected neurons is approximately 2,000-3,750/well. However, what also needs to be factored in here is the consequence of superinfection itself. Neurons are much more repressive for lytic infection; most studies thus far using superinfection have been performed in fibroblasts to show re-expression of lytic transcripts from quiescent, replication defective viral genomes. Using the same MOI with a reporter virus that is unable to spread cell-to-cell, we have found that the virus undergoes lytic replication in only a fraction of neurons.

There are many implications for this low induction of reactivation, which may relate to the heterogeneity of the neuron themselves, differences in the epigenetic structure of the viral genome from one neuron to another or even communication between cells that limits reactivation from neighboring cells. We are currently carrying out studies investigating the outcome of neuronal infection at the single cell basis. There is much work to be done that is beyond the scope of this publication.